# 🪆 MatryoshkaKV: Adaptive KV Compression via Trainable Orthogonal Projection

**Bokai Lin[1]  Zihao Zeng[1]  Zipeng Xiao[1]  Siqi Kou[1]**
**Tianqi Hou[2]  Xiaofeng Gao[1]  Hao Zhang[3]  Zhijie Deng[1]***
[1]Shanghai Jiao Tong University [2]Huawei [3]University of California, San Diego
`{19821172068,zengzihao,xiaozp_25,happy-karry,zhijied}@sjtu.edu.cn`
`thou@connect.ust.hk, gao-xf@cs.sjtu.edu.cn, haozhang@ucsd.edu`

## Abstract

KV cache has become a *de facto* technique for the inference of large language models (LLMs), where tensors of shape (layer number, head number, sequence length, feature dimension) are introduced to cache historical information for self-attention. As the size of the model and data grows, the KV cache can quickly become a bottleneck within the system in both storage and memory transfer. To address this, prior studies usually focus on the first three axes of the cache tensors for compression. This paper supplements them, focusing on the feature dimension axis, by utilizing low-rank projection matrices to transform the cache features into spaces with reduced dimensions. We begin by investigating the canonical orthogonal projection method for data compression through principal component analysis (PCA). We observe the issue with PCA projection where significant performance degradation is observed at low compression rates. To bridge the gap, we propose to directly tune the orthogonal projection matrices with a distillation objective using an elaborate Matryoshka training strategy. After training, we adaptively search for the optimal compression rates for various layers and heads given varying compression budgets. Compared to previous works, our method can easily embrace pre-trained LLMs and hold a smooth tradeoff between performance and compression rate. We empirically witness the high data efficiency of our training procedure and find that our method can sustain over 90% performance with an average KV cache compression rate of 60% (and up to 75% in certain extreme scenarios) for popular LLMs like LLaMA2-7B-base and Mistral-7B-v0.3-base.

## 1 Introduction

Large language models (LLMs) like GPT-4 (OpenAI et al., 2024) and Claude3 (Enis & Hopkins, 2024) have shown great promise, finding applications in areas such as text generation (Brown et al., 2020; Raffel et al., 2023), code completion (Rozière et al., 2024), and sentiment analysis (Zhang et al., 2023a). The Key-Value (KV) cache, which is introduced to cache historical information for self-attention, is essential for maintaining context and accelerating the inference of LLMs. However, as the size of the model and data continues to grow (Fu et al., 2024; Ding et al., 2024; Chen et al., 2024), the KV cache can swiftly lead to system bottleneck in terms of storage and memory transfer (Shi et al., 2024).

Considerable efforts have been devoted to addressing such an issue. Noting that the KV cache contains tensors of shape (layer number, head number, sequence length, feature dimension), existing works have investigated compressing the KV cache from the axes of layer number (Brandon et al., 2024; Sun et al., 2024; Goldstein et al., 2024), head number (Ainslie et al., 2023; Shazeer, 2019; Yu et al., 2024), and sequence length (Wang et al., 2024; Zhang et al., 2023b; Li et al., 2024; Xiao et al., 2024). Conversely, the exploration of feature dimension for KV cache compression significantly lags behind, partially because of the inherent difficulties of modifying a well-structured feature space.

---

*Corresponding author.

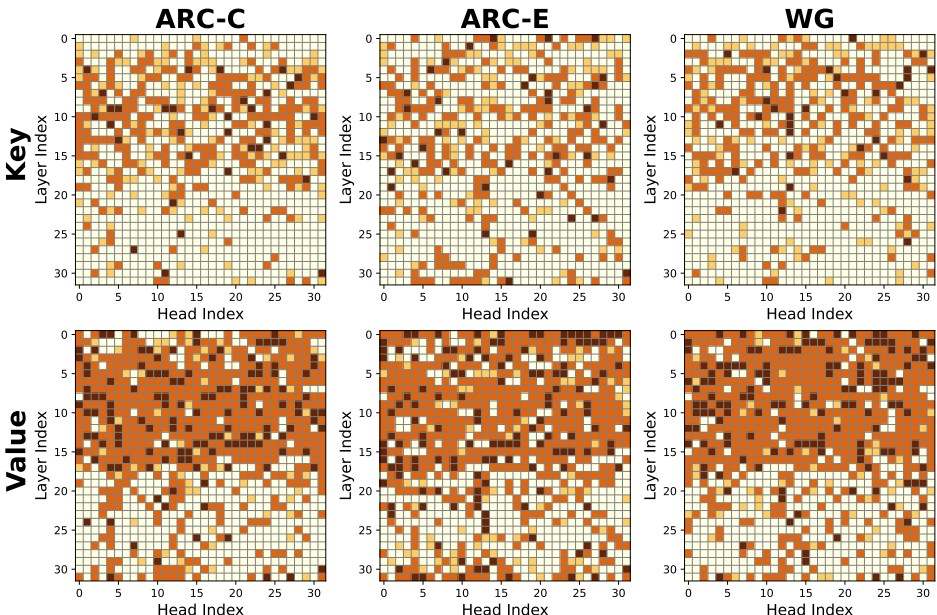

Figure 1: Visualization of the feasible compression level for the keys and values in our model distilled from the LLaMA2-7B-base model. We individually leverage samples in ARC-challenge (ARC-C), ARC-easy (ARC-E) (Clark et al., 2018), and Winogrande (WG) (Sakaguchi et al., 2019) to determine the compression level. Lighter colors indicate higher compression levels. As shown, our approach enables the use of various compression strategies for various tasks.

This paper aims to tackle this with the help of curated low-rank projection matrices, e.g., both the query and key are projected into the same lower-dimensional space wherein the inner product closely approximates that in the original space. We first identify the necessity to guarantee the orthogonality among the rows of such matrices, and hence attempt to take the principal components of the keys or values in each layer to instantiate the projections, given the prevalence of Principal Component Analysis (PCA) for data compression. We observe that such projections can be seamlessly plugged into pre-trained LLMs while retaining reliable generation quality at a moderate compression level. Compared to the low-rank architectures of Multi-head Latent Attention (MLA) (DeepSeek-AI et al., 2024), the PCA strategy is more approachable due to its training-free nature and also advocated by Saxena et al. (2024). Yet, we note that the PCA projections suffer from quickly degraded performance when further increasing the compression level. This is because, while the principal components are optimal for recovering the keys or values in each individual layer, they may be suboptimal for preserving the global outputs due to the non-linearity and compounding effects in LLM.

To bridge the gap, we propose to jointly adjust all orthogonal projection matrices incorporated into the model with a knowledge distillation objective, enforcing the model output based on the projected keys and values to remain close to the original one. The orthogonality constraint upon the projection matrices is consistently enforced by a Cayley parameterization. Besides, we desire a hierarchy over the columns of the projection matrices—as in PCA—so that we can smoothly trade-off between compression level and performance. To this end, we introduce a Matryoshka training strategy—compute the model output based on the first $r$ columns of the matrices, where $r$ is randomly sampled from a predefined schedule such as $\{4, 8, 16, ...\}$, and ensure its closeness to the original output. In practice, we sample various $r$ for different layers, heads, and keys/values during training to disentangle the projections in the model. Doing so enables the search for heterogeneous compression rates for different projection matrices during inference and we develop a greedy algorithm for this. Heterogeneous compression rates are displayed in Figure 1.

Experiments on both continual pre-training (CPT) and supervised fine-tuning (SFT) exhibit the efficacy of our MatryoshkaKV approach. For the former, we opt to experiment on LLaMA2-7B-base (Touvron et al., 2023) with the RedPajama dataset (Computer, 2023). To demonstrate compatibility with Group Query Attention (GQA) (Ainslie et al., 2023), we also apply our approach to the Mistral-v0.3-7B-base (Jiang et al., 2023) model. Moreover, we demonstrate that MatryoshkaKV

is compatible with other KV cache compression techniques on other axes like H$_2$O (Zhang et al., 2023b) and KIVI (Liu et al., 2023). We observe that after rarely processing 200 million training tokens, MatryoshkaKV achieves a 37.5% compression rate while retaining over 90% of the original model's accuracy. In the SFT experiments, we train both MatryoshkaKV and LoRA (Hu et al., 2021) on downstream tasks including OBQA (Mihaylov et al., 2018), GSM8K (Cobbe et al., 2021), etc. The results show that our MatryoshkaKV can utilize less than 40% cache while still achieving over 90% accuracy derived from full cache utilization. We also perform extensive ablation studies to chase a deep understanding of our approach. The code is available at `https://github.com/The-kamisato/MatryoshkaKV-cache.git`.

## 2 RELATED WORK

**KV cache eviction & merging.** KVMerger (Wang et al., 2024) and PyramidKV (Cai. et al., 2024) introduce innovative approaches to reduce KV cache memory consumption along sequence length dimension in long-context tasks. KVMerger merges KV by Gaussian weights and attention score, while PyramidKV uses a layer-wise approach with recent tokens occupying more weights. CLA (Brandon et al., 2024), YOCO (Sun et al., 2024), and GoldFinch (Goldstein et al., 2024), among others, exploit inter-layer KV cache reuse by sharing KV heads across layers. This significantly reduces the KV cache size along the head number dimension without compromising model capacity. GQA (Ainslie et al., 2023), MQA (Shazeer, 2019), and HeadKV Yu et al. (2024), especially the last one, have demonstrated the effectiveness of compressing KV cache on the axis of head number due to their low-rank properties.

**KV cache hidden size compression.** DeepSeekv2 (DeepSeek-AI et al., 2024) employs MLA techniques to reduce the feature dimension of keys and values within the attention mechanism, but this requires costly retraining from scratch. Concurrent advancements, however, have addressed this limitation. Eigen-Attention (Saxena et al., 2024) and HeadKV (Yu et al., 2024) achieve a 40% reduction in the KV cache sizes using orthogonal projections parameterized by the SVD of the Q, K, and V matrices derived from a subset of samples. To mitigate performance degradation, LoRA (Hu et al., 2021) is employed to fine-tune model parameters. However, this compression approach on the axis of feature dimension results in a sharp decline in model performance when using less than 60% cache budget. Furthermore, fine-tuning the base model with LoRA may lead to catastrophic forgetting. In this paper, our method MatryoshkaKV circumvents these risks and achieves higher compression rate by directly fine-tuning orthogonal projections.

## 3 PRELIMINARY

This section provides a review of the KV cache mechanism and elucidates the implementation of PCA projection for KV cache compression.

### 3.1 KV CACHE

Consider the inference of an LLM $p(\cdot|\boldsymbol{x})$ with $\boldsymbol{x}$ as the prompt. It is a common practice to deploy the KV cache technique to each self-attention head in the model to store the key and value states for the present context, including both the prompt $\boldsymbol{x}$ and the tokens that have already been generated. Given the KV cache for the context of length $L-1$ and dimension $d$ in each head, the model generates a subsequent new token $y$ with the attention states $\text{softmax}(QK^\top/\sqrt{d})V$, where $Q \in \mathbb{R}^{1\times d}$ is the query vector for $y$ and $K, V \in \mathbb{R}^{L\times d}$ denote the concatenation of the KV cache and the KV vectors for $y$. This way, the computational complexity for one decoding step is reduced from $\mathcal{O}(L)$ to $\mathcal{O}(1)$.

However, the size of the KV cache can grow quickly w.r.t. that of the model and context, often causing system bottlenecks in terms of both storage and memory transfer during the inference phase. To address this, various KV cache compression techniques have been proposed, e.g., sharing the KV headers across layers inside LLMs (Brandon et al., 2024; Sun et al., 2024; Goldstein et al., 2024), merging heads that require caching KV (Yu et al., 2024; Ainslie et al., 2023), evicting or merging redundant tokens (Xiao et al., 2024; Li et al., 2024; Cai. et al., 2024; Zhang et al., 2023b). This work alternatively focuses on compressing the feature dimension $d$ of the KV cache, exploring a novel axis for KV cache compression that is compatible with existing methodologies.

## 3.2 Traing-free Dimension Reduction Via PCA

A simple way to reduce the dimension of the KV cache is finding some matrices to project $K, V$ as $K', V' \in \mathbb{R}^{L \times r}, (r < d)$. Then, we can only cache $K'$ and $V'$, reducing the storage and memory transfer cost from $\mathcal{O}(d)$ to $\mathcal{O}(r)$. The rank $r$ is desired to be adjustable based on the available compression budget: when the budget is sufficient, caching full KV states helps prevent information loss; in cases of limited budget, caching only the most essential information should be feasible. To fulfill this, it is reasonable to introduce full-rank projection matrices $U \in \mathbb{R}^{d \times d}$ and demand a hierarchy over the columns of $U$ so that the optimal $r$-rank cache can result from the first $r$ columns of $U$, denoted as $U_r \in \mathbb{R}^{d \times r}$. In practice, $U$ should be distinct for $K$ and $V$ and vary across attention heads and layers within the model, as these states commonly exhibit diverse distributions.

During the forward pass of the model, we should be able to recover the original $K$ and $V$ from the reduced $K'$ and $V'$. A natural choice is using $U^\top$, the transposition of the projection matrices, where $U_r U_r^\top \approx I$ needs to be satisfied. Given that $r$ can vary from 1 to $d$, we identify that $U$ should be orthogonal matrices. It is known that the optimal orthogonal projections for compressing a set of high-dimension vectors can be their principal components, so we suggest constructing $U$ based on the PCA results of the key or value states of a long sequence of tokens for each head separately.

Table 1 displays an empirical study of the efficacy of such training-free projections. As shown, PCA projections exhibit reliable performance at moderate levels of compression budget. This is remarkable because the PCA strategy does not need costly from-scratch training of the projection matrices, in sharp contrast to the projection mechanisms used by MLA (DeepSeek-AI et al., 2024). We note that PCA projection is also advocated by Saxena et al. (2024); refer to Appendix B for the difference between our attempts and theirs regarding applying projections before or after RoPE (Su et al., 2023) and whether performing fine-tuning.

Nevertheless, as the table displays, the PCA projections suffer from quickly degraded performance when further increasing the compression level. This is because despite principal components being optimal for key or value recovery in individual head layers, they may be inadequate for preserving the final output due to the non-linearity and compounding effects of the attention mechanism.

## 4 Methodology

To address the aforementioned issue, we propose to jointly tune the orthogonal projection matrices introduced to the LLM under an elaborate objective, to realize a more robust KV cache compression. The whole pipeline can be listed as follows: (1) Obtain the PCA initialization based on a small subset of a general corpus. (2) Train our model on the corpus. (3) Search for the heterogeneous compression levels for various heads with a small calibration dataset (5 - 10 samples) on the specific task. (4) Perform inference on that task given the identified compression levels. This section provides the training and inference details of our approach.

### 4.1 Minimize Compression Loss by Knowledge Distillation

Recalling the objective for the compression is that the model outputs based on the compressed states should stay close to the original one. This implies a knowledge distillation objective (Hinton et al., 2015), which can be instantiated with the KL divergence:

$$\mathcal{L}_{KD} = D_{\mathrm{KL}}(p\left(\cdot|\boldsymbol{x}\right) \| p'\left(\cdot|\boldsymbol{x}\right)) \tag{1}$$

where we abuse $p'$ to refer to the LLM equipped with low-rank projection matrices. As suggested by the literature (Kou et al., 2024), we also incorporate a language modeling loss to $p'$ to prevent the generated text from deviating from the context distribution of the dataset, thereby ensuring high-quality generation. The tuning process involves only the update of $U$, which ensures that the model performance under the full-rank KV cache is maintained.

**Orthogonal constraint.** We initialize the trainable orthogonal projections with the PCA ones due to their effectiveness. To confine the evolution of the projection matrices within the orthogonal matrix family throughout the tuning process, we employ Cayley parameterization to formulate the orthogonal matrix. Specifically, there is $U = (I + Q)(I - Q)^{-1}$ with $Q$ as a skew-symmetric

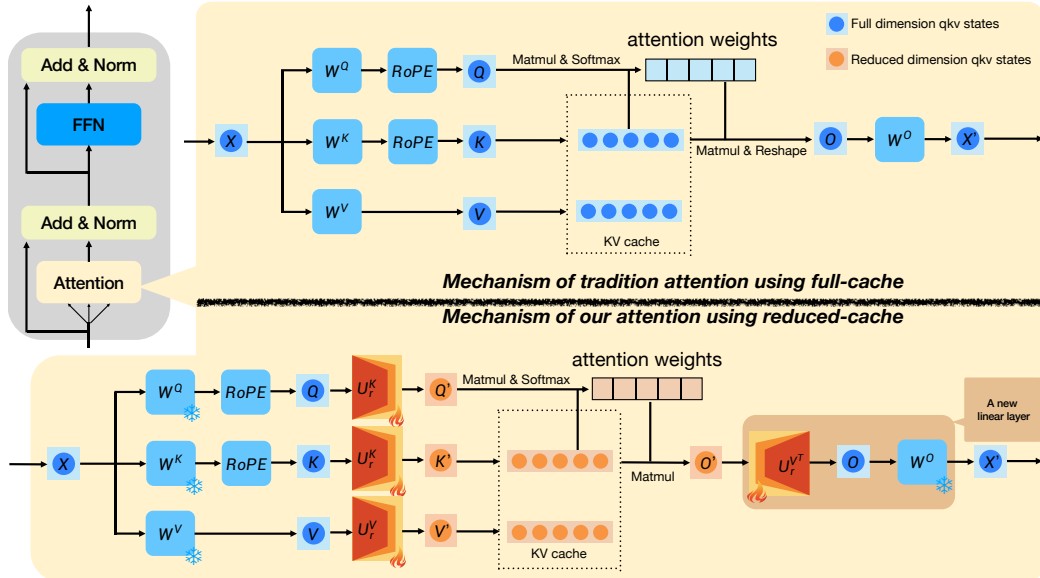

Figure 2: Vanilla KV cache vs. the proposed MatryoshkaKV. In particular, we introduce orthogonal projection matrices to reduce the dimension of stored keys and values. We explicitly enforce a hierarchy over the columns of projection matrices so as to concentrate the principal information on the head dimensions and enable the adjustment of compression level according to resource constraints.

trainable matrix of size $d \times d$. Considering that $d$ is usually small (e.g., 64 or 128), the complexity of performing such an orthogonal transformation during training is minimal.

## 4.2 ACQUIRE HIERARCHICAL KV CACHE BY MATRYOSHKAKV TRAINING

The tuning process can destroy the hierarchical structures present in the orthogonal matrices inherited from the PCA ones because there is no prioritization given to the columns of the matrices $U$ from the training objective. Consequently, we lose the flexibility to achieve a smooth transition between the level of compression and maintenance of the original performance.

To tackle this challenge, we draw inspiration for Matryoshka representation learning (Kusupati et al., 2022), introducing a Matryoshka strategy for training the projection matrices $U$. In particular, for each training iteration, we randomly sample $r$ from a predefined schedule such as $\{4, 8, 16, ..., d/4, d/2, d\}$ and use the first $r$ columns of $U$, i.e., $U_r$, to construct the model $p'$ for training. Note that the keys and values at different heads and layers use separately sampled $r$ to avoid the entanglement of the compression effect. An illustrative explanation of this is given in Figure 4, and our approach is then called MatryoshkaKV for short.

## 4.3 FIND HETEROGENEOUS COMPRESSION RATES FOR VARIOUS LAYERS & HEADS

The Matryoshka training strategy enables the search for heterogeneous compression rates for various layers and heads in the model given a specific compression budget. Basically, we can first propose a compression level for the projection matrix at a particular position, assessing the deviation of the model output from the original on a predefined calibration dataset (measured by KL divergence), and determining whether to accept the proposal based on a predefined tolerance threshold for the deviation. Algorithm 1 exhibits a greedy algorithm for accelerating this based on accepting proposals in parallel. Note that this greedy algorithm also applies to the PCA projections.

**Discussion.** The recent KV cache compression approach on sequence length aspect (Cai. et al., 2024) also observes that compared to uniformly compressed KV cache using the same rate across all layers (Li et al., 2024), employing a distinct compression rate for each layer results in improved information utilization. Furthermore, as observed in (Wu et al., 2024), certain retrieval heads within

---

**Algorithm 1:** Greedy search for adaptive compression levels in our efficient LLM.

---

**input :** An base LLM $p\left(\cdot\right)$ and an efficient LLM equipped with MatryoshkaKV projections $p'\left(\cdot\right)$, layer num $L$, attention head num $H$, full KV cache feature dimension $d$, a prompt $\boldsymbol{x}$, compression rate interval $\Delta r$, target cache budget $\gamma$.

**output:** Two tensors $R^K, R^V \in \mathbb{R}^{L \times H}$ specifying the heterogeneous key/value compression rates for each head in each layer.

$R^K, R^V \leftarrow d \cdot \mathbb{1}^{L \times H}$

**repeat**

    $R_{temp}^K, R_{temp}^V \leftarrow R^K, R^V$

    **for** *Every Layer-l in LLM* **do**

        **for** *Every Attention Head-h* **do**

            $R_{temp,l,h}^K, R_{temp,l,h}^V \leftarrow R_{l,h}^K - \Delta r, R_{l,h}^V - \Delta r$

            $\varepsilon_{l,h}^K \leftarrow D_{\mathrm{KL}}\left(p\left(\cdot|\boldsymbol{x}\right)\|p'\left(\cdot|\boldsymbol{x}; R_{temp}^K, R^V\right)\right)$

            $\varepsilon_{l,h}^V \leftarrow D_{\mathrm{KL}}\left(p\left(\cdot|\boldsymbol{x}\right)\|p'\left(\cdot|\boldsymbol{x}; R^K, R_{temp}^V\right)\right)$

            $R_{temp,l,h}^K, R_{temp,l,h}^V \leftarrow R_{l,h}^K, R_{l,h}^V$

    Locate the index associated with the minimum value element in the joint error list $[\varepsilon^K, \varepsilon^V]$.

    Decrement the corresponding compression rate in $[R^K, R^V]$ by $\Delta r$.

**until** *Budget* $\left(R^K, R^V\right) < \gamma$;

**return** $R^K, R^V$

---

an LLM consistently attend to crucial information, regardless of contextual variations. The indiscriminate compression rates of these heads can lead to significant performance degradation. These both support the necessity of the proposed heterogeneous KV cache compression approach.

## 5   EXPERIMENTS

In this section, we conduct experiments on continual pre-training (CPT) and supervised fine-tuning (SFT) scenarios to demonstrate that our MatryoshkaKV can not only preserve the foundation knowledge of a base model but also be compatible with LoRA (Hu et al., 2021) for downstream tasks. Furthermore, we combine our approach with a KV cache compression technique targeting the sequence length dimension, referred to as H$_2$O (Zhang et al., 2023b), and additionally implement another experiment by integrating KIVI (Liu et al., 2023), a KV cache compression strategy focused on KV cache quantization. Ablation studies in Section 5.4 validate the efficacy of our proposed method.

### 5.1   CONTINUAL PRE-TRAINING

**Setup.** We select LLaMA2-7B-base (Touvron et al., 2023) and Mistral-v0.3-7B-base (Jiang et al., 2023) as our base models. We conduct continual pre-training (Ke et al., 2023) using the RedPajama dataset (Computer, 2023). To rapidly validate the effectiveness of our proposed method, we choose a subset of this dataset following RedPajama-Data-1T-Sample. We adopt the Matryoshka training strategy detailed in Section 4.2 and fine-tune MatryoshkaKV projections with knowledge distillation loss in Equation 1 and language modeling loss, applying a 1:3 weighting ratio between the two losses. The projection rank $r_k$ and $r_v$ are randomly sampled from a predefined schedule set $\{\frac{i}{8}d\}_{i=1}^8$ during training and are chosen dynamically with the greedy search for adaptive compression levels, as detailed in Section 4.3 during inference. During the greedy search for adaptive compression levels, we define the compression rate interval $\Delta r = d/8$ where the head dimension $d$ for each attention head in LLaMA2-7B-base is 128. We use Opencompass (Contributors, 2023) to test performance on several widely-used zero-shot benchmarks: PIQA (Bisk et al., 2019), ARC-challenge (ARC-C) (Clark et al., 2018), ARC-easy (ARC-E) (Clark et al., 2018), WinoGrande (WG) (Sakaguchi et al., 2019), HellaSwag (HLSG) (Zellers et al., 2019), and CommonSenseQA (CSQA) (Talmor et al., 2019). We compare our methods with Eigen-attention (Saxena et al., 2024) (donated as PCA) in Table 1 and ASVD (Yuan et al., 2024) in Table 7.

Table 1: Comparison between our MatryoshkaKV method (donated as MKV in the table) and PCA projection. We use LLaMA2-7B-base and Mistral-v0.3-7B-base as our source models, and their performance is used as a baseline. Accuracy on HellaSwag, ARC-challenge, ARC-easy, PIQA, Wino-Grande, and CommonSenseQA is reported, with higher scores indicating superior performance, at seven KV cache budgets. At the same budget, the higher average accuracy is underlined.

| Model | Budget | Method | HLSG | ARCC | ARCE | PIQA | WG | CSQA | Avg. |
|---|---|---|---|---|---|---|---|---|---|
| LLaMA2 7B-base | 100.0% | Baseline | 74.00 | 35.93 | 50.97 | 78.50 | 61.64 | 65.93 | 61.16 |
| | | PCA | 72.04 | 36.95 | 52.38 | 76.66 | 61.72 | 67.24 | 61.17 |
| | | MKV | 72.05 | 37.29 | 52.38 | 76.66 | 61.72 | 67.32 | 61.24 |
| | 87.5% | PCA | 71.91 | 35.93 | 53.97 | 76.66 | 61.40 | 67.65 | 61.25 |
| | | MKV | 71.58 | 37.97 | 53.26 | 75.95 | 62.12 | 69.57 | 61.74 |
| | 75.0% | PCA | 70.99 | 35.59 | 54.14 | 76.22 | 60.06 | 66.99 | 60.67 |
| | | MKV | 71.58 | 38.31 | 55.56 | 76.01 | 61.09 | 66.75 | 61.55 |
| | 62.5% | PCA | 67.16 | 34.24 | 54.85 | 74.76 | 57.77 | 61.10 | 58.31 |
| | | MKV | 68.03 | 37.97 | 56.08 | 75.12 | 60.30 | 65.44 | 60.49 |
| | 50.0% | PCA | 42.11 | 29.83 | 35.10 | 58.16 | 52.57 | 40.62 | 43.07 |
| | | MKV | 66.78 | 36.61 | 55.91 | 74.32 | 59.12 | 61.92 | 59.11 |
| | 37.5% | PCA | 24.24 | 26.44 | 26.63 | 51.25 | 50.36 | 19.90 | 33.14 |
| | | MKV | 63.97 | 33.90 | 51.68 | 74.97 | 57.92 | 59.21 | 56.94 |
| | 25.0% | PCA | 23.98 | 29.49 | 26.28 | 51.20 | 50.36 | 16.22 | 32.92 |
| | | MKV | 51.91 | 27.46 | 44.44 | 69.64 | 54.54 | 44.39 | 48.73 |
| Mistral-v0.3 7B-base | 100.0% | Baseline | 75.50 | 42.03 | 63.14 | 80.25 | 65.43 | 70.68 | 66.17 |
| | | PCA | 75.46 | 42.03 | 62.96 | 80.25 | 65.35 | 70.27 | 66.05 |
| | | MKV | 75.44 | 42.03 | 62.96 | 80.25 | 65.51 | 70.27 | 66.08 |
| | 87.5% | PCA | 73.46 | 42.71 | 63.32 | 79.54 | 63.93 | 70.76 | 65.92 |
| | | MKV | 75.63 | 42.03 | 64.37 | 79.71 | 65.51 | 70.35 | 66.27 |
| | 75.0% | PCA | 70.75 | 37.63 | 61.73 | 78.18 | 62.59 | 68.47 | 63.23 |
| | | MKV | 75.29 | 43.39 | 63.14 | 79.54 | 64.96 | 69.12 | 65.90 |
| | 62.5% | PCA | 63.48 | 34.24 | 55.73 | 75.90 | 60.77 | 62.24 | 58.73 |
| | | MKV | 74.23 | 40.34 | 62.96 | 79.33 | 64.25 | 68.63 | 64.96 |
| | 50.0% | PCA | 28.12 | 22.71 | 28.40 | 58.16 | 49.64 | 22.85 | 34.98 |
| | | MKV | 73.32 | 38.98 | 62.08 | 79.16 | 61.88 | 67.08 | 63.75 |
| | 37.5% | PCA | 25.04 | 22.03 | 28.04 | 53.86 | 49.25 | 21.21 | 33.24 |
| | | MKV | 70.40 | 35.93 | 58.91 | 77.91 | 60.30 | 64.29 | 61.29 |
| | 25.0% | PCA | 24.91 | 26.10 | 25.40 | 52.67 | 48.30 | 19.74 | 32.85 |
| | | MKV | 59.21 | 25.42 | 48.68 | 73.83 | 54.30 | 45.13 | 51.10 |

**Results.** We train with a total of 30 GPU × hours, processing just under 200 million tokens (20% of the RedPajama sample 1T, i.e. 0.02% of the full RedPajama dataset). Table 1 presents the results of our experiments. In zero-shot tasks, our MatryoshkaKV cache substantially reduces the cache footprint with minimal impact on performance. Specifically, our method retains 93.10% of LLaMA2-7B-base's average accuracy and 92.63% of Mistral-v0.3-7B-base's average accuracy, while utilizing only 37.5% of the original cache size. For simpler tasks like PIQA, it achieves 88.71% and 92.00% of the base model's performance with just a 25% cache budget. On more challenging tasks such as ARC-C, a larger cache budget is required, with 50% needed to retain 90% of the base model's performance. By contrast, PCA projection shows a sharp performance drop when the cache budget is reduced below 62.5%, achieving just 70.42% accuracy of LLaMA2-7B-base and 52.86% of Mistral-v0.3-7B-base. These results underscore the superior efficiency of our approach compared with PCA. We attribute PCA's performance decline to suboptimal projection matrices, whereas our method maintains closer alignment with the base model, thereby mitigating this degradation.

Furthermore, we evaluate the inference speed of our LLM equipped with our MatryoshkaKV. The results are presented in Table 3 and related discussions are detailed in Appendix F.

Table 2: Accuracy of our Matryoshka method after SFT based on LLaMA2-7B-base on four downstream tasks: PIQA, GSM8K, HellaSwag, and OpenbookQA, at seven KV cache budgets. Degradation from baseline is shown in brackets.

| Model | Budget | PIQA | GSM8K | HLSG | OBQA | Avg. |
|-------|--------|------|-------|------|------|------|
| LLaMA2 7B-base | 100.0 % | 84.22 | 34.95 | 93.94 | 83.2 | 74.08 (-0.00%) |
| | 87.5 % | 83.84 | 35.25 | 93.17 | 81.80 | 73.52 (-0.76%) |
| | 75.0 % | 83.30 | 32.90 | 91.47 | 81.40 | 72.27 (-2.44%) |
| | 62.5 % | 82.75 | 31.46 | 89.86 | 79.60 | 70.92 (-4.27%) |
| | 50.0 % | 79.33 | 31.77 | 86.29 | 76.60 | 68.50 (-7.53%) |
| | 37.5 % | 75.35 | 26.91 | 76.10 | 70.80 | 62.29 (-15.9%) |
| | 25.0 % | 69.04 | 16.38 | 56.10 | 61.40 | 50.73 (-31.5%) |

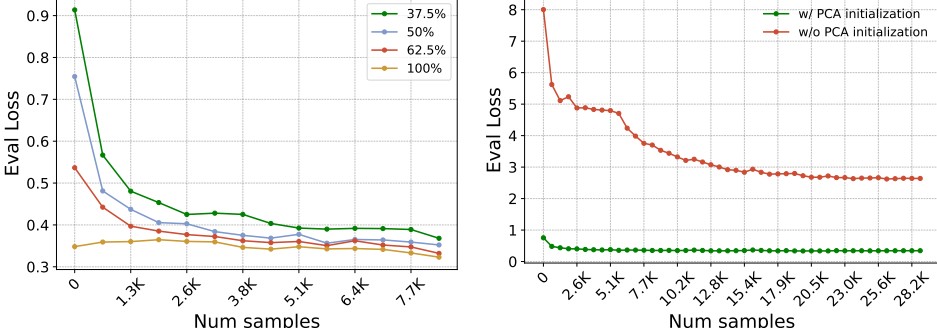

Figure 3: Evaluation loss of four budgets vs. the number of training samples during 1 epoch of SFT on GSM8K (Left). Evaluation loss of models with and without PCA initialization, using a 50% cache budget, vs. the number of training samples during 4 epochs of SFT on GSM8K (Right).

## 5.2 SUPERVISED FINE-TUNING

**Setup.** We use LLaMA2-7B-base (Touvron et al., 2023) as our base model and verify the efficacy of our method on PIQA (Bisk et al., 2019), GSM8K (Cobbe et al., 2021), HellaSwag (Zellers et al., 2019), and OpenbookQA (OBQA) (Mihaylov et al., 2018) datasets. We design a two-stage training strategy to make Matryoshka training strategy compatible with LoRA (Hu et al., 2021) fine-tuning. Specifically, LoRA is firstly used to adapt the base model to downstream tasks, following standard SFT practices (Naveed et al., 2024; Zhao et al., 2024). In the second stage, we jointly fine-tune the MatryoshkaKV projections with the Matryoshka training strategy and the LoRA parameters. Further discussion on the superiority of this recipe is detailed in Appendix C.

**Results.** We report accuracy on four zero-shot benchmarks at seven KV cache budgets in Table 2. As shown, our method demonstrates notable performance in the SFT scenario. It achieves 92.47% of the baseline's average accuracy while utilizing only 50% of the KV cache budget. On simple tasks like PIQA, our method retains 89.47% of the full-cache performance with a 37.5% cache budget. However, for more complex tasks such as GSM8K, a 50% cache budget is necessary to achieve comparable results. Furthermore, we report the evaluation loss at four budgets: 100%, 62.5%, 50%, and 37.5% during the second stage of SFT on GSM8K in Figure 3 (Left). It shows our method simultaneously optimizes models under various KV cache budgets and maintains the hierarchical structures present in the orthogonal matrices. These findings highlight the robustness of our approach, delivering consistent performance across both CPT and SFT scenarios.

## 5.3 COMPATIBILITY WITH OTHER KV CACHE COMPRESSION TECHNIQUES

To demonstrate the orthogonality and compatibility of our method with existing KV cache compression techniques, we conduct extensive experiments utilizing MatryoshkaKV in conjunction with these methods. Based on the classification outlined in Section 2, we integrate MatryoshkaKV

Table 3: Tokens per second at different KV cache budgets with a batch size of 32.

| | LLaMA2 | 100% | 87.5% | 75% | 62.5% | 50% | 37.5% | 25% |
|---|---|---|---|---|---|---|---|---|
| Tokens per second | 33.65 | 34.12 | 34.08 | 34.90 | 35.27 | 36.42 | 36.75 | 37.22 |

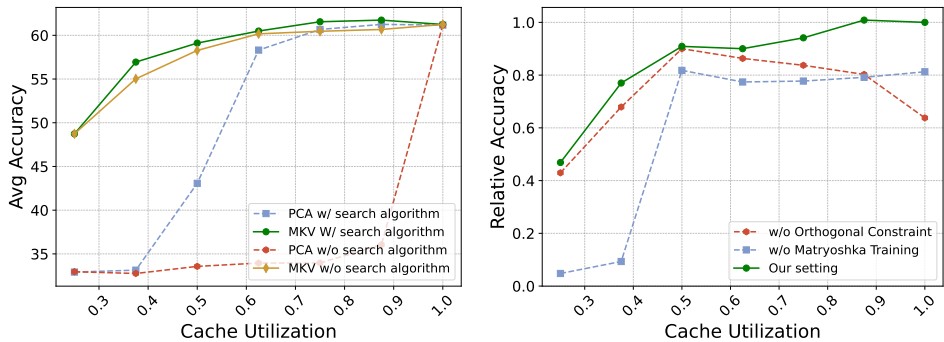

Figure 4: Comparison between PCA and distilled MatryoshkaKV Projections after CPT with and without greedy search for adaptive compression levels. We report average accuracy on datasets mentioned in the experimental setup of Section 5.1 (Left). Comparison between with and without Matryoshka training strategy and orthogonal constraint after SFT on GSM8K. We report the relative accuracy compared with the LLaMA2-7B-base model fine-tuned with LoRA on GSM8K, utilizing the full KV cache (Right).

with prominent techniques such as KIVI (Hooper et al., 2024) for KV quantization, $H_2O$ (Zhang et al., 2023b), and GQA (Ainslie et al., 2023) for KV cache eviction and merging. We apply MatryoshkaKV to Mistral-v0.3-7B-base in Section 5.1, demonstrating its enhanced compression capability in synergy with GQA (Ainslie et al., 2023).

**Combination with $H_2O$.** Furthermore, we combine our methods with $H_2O$. We first evaluate $H_2O$ and our MatryoshkaKV on datasets mentioned in 5.1. To demonstrate improved compression rates in long contexts, we select LongBench (Bai et al., 2024) and calculate perplexity under different cache budget settings of two orthogonal KV compression techniques. For the detailed results, please refer to Table 5 in Appendix E.

According to the results, by concurrently using MatryoshkaKV and $H_2O$, the perplexity on long contexts increases by merely 1.02 at 10% KV cache budget. Additionally, if we compress by 50% on both the sequence length and feature dimension axes (with an actual cache usage rate of 25%), we can achieve an average accuracy of 55.85 on 6 benchmarks, which is 91.32% of the baseline.

**Combination with KIVI.** In addition to integrating with $H_2O$, we also explore the combination of our methods with KIVI (Liu et al., 2023), a KV cache compression technique based on 2-bit cache quantization. Similar to the previous approach, we conduct evaluations on the datasets described in Section 5.1. The detailed results of this combination are presented in Table 6 in Appendix E and analyzed in detail. The results show that our MatryohskaKV can be easily combined with KV quantization techniques and achieve a higher compression rate.

## 5.4 ABLATION STUDIES

We conduct ablation studies on various components of our method to verify their effectiveness.

**W/o greedy search for adaptive compression levels.** We evaluate our trained models without our greedy search for adaptive compression levels. Figure 4 (Left) represents the average accuracy on four datasets mentioned in Section 5.2 as the cache budget varies. For the exact numerical values, please refer to Table 4 in Appendix D. To ensure that each head in the LLM plays its due role, we set 25% as our minimum cache budget for each head. At a 37.5% cache budget, the average accuracy improves by 1.92%, indicating the significance of our search algorithm for further KV cache compression. Furthermore, our MatryoshkaKV demonstrates robustness even when applying

a uniform compression rate across all layers and heads, in contrast to PCA projection, which fails to handle this setting effectively.

**W/o Matryoshka training strategy.** As discussed in Section 4.2, we point out that the tuning process w/o Matryoshka training strategy can destroy the hierarchical structures present in the orthogonal matrices inherited from the PCA ones. To validate this, we train MatryoshkaKV projections with a fixed KV cache budget of 50%. The result is displayed in Figure 4 (Right). We observe that fixing the compression rate at 50% hinders the potential for further compression. Moreover, when the budget exceeds 50%, the model's performance does not improve significantly but even deteriorates, indicating the hierarchical structure of projections is destroyed.

**W/o orthogonal constraint.** We investigate the necessity of imposing the orthogonal constraint during training, with experimental results presented by Figure 4 (Right). After training without orthogonal constraint on GSM8K, we observe that non-orthogonal projections achieve performance comparable to orthogonal projections when the cache budget is less than 50%. However, when utilizing a full KV cache budget, this model is unable to maintain the performance of the base model. This is due to the non-orthogonality of the projection matrix, which prevents LLM from replicating the attention mechanism of the base model. This phenomenon also validates our discussion in previous Section 3.2.

**W/o PCA initialization.** To demonstrate the necessity of using PCA results to initialize projections, we train an LLM equipped with randomly initialized orthogonal matrices on GSM8K and impose orthogonal constraints. In Figure 3 (Right), we report the evaluation loss during the second stage of SFT on GSM8K. Despite training for four epochs, randomly initialized orthogonal projections fail to converge to an optimal solution, and the text generated by our fine-tuned LLM projection is composed of meaningless symbols. This highlights the importance of PCA initialization.

## 5.5 HETEROGENEOUS COMPRESSION RATES VISUALIZATION

Figure 1 shows the heterogeneous compression levels across all attention heads inside our MatryoshkaKV LLM distilled from the LLaMA2-7B-base. We acquire these results by leveraging the greedy search for adaptive compression levels on the ARC-C, ARC-E, and WinoGrande datasets. We observe that shallower layers require larger KV cache budgets, while in deeper layers, only a minority of specific heads require a relatively high budget. PyramidKV (Cai. et al., 2024) also observes that the model aggregates information globally from all available content in lower layers, indicating that KV cache inside lower layers can exert a substantial influence over the final output and should be allocated at a relatively high budget. Therefore, allocating more cache in lower layers and less in higher ones is superior to maintaining a uniform KV cache size across layers. Also, as Wu et al. (2024) point out, retrieval heads with high retrieval scores in LLaMA2-7B-base, are much more important than other heads and should be preserved in KV cache compression. These findings are consistent with our observations.

Moreover, we observe that keys can be more compressed than values. As shown by the heatmaps in Appendix D, the compression of values affects downstream tasks more than keys. Specifically, according to our greedy search for adaptive compression levels, for a 37.5% KV cache budget, the optimized key cache budget is allocated 32.28%, and the value cache budget is allocated 42.72%.

## 6 CONCLUSION

In this study, we delve into how to compress the KV cache in LLMs by applying low-rank projection matrices to the feature dimension. We first investigate data compression using the canonical orthogonal projection method through PCA. We observe significant performance degradation at a relatively high compression rate, indicating that PCA projection is suboptimal for preserving global outputs due to LLMs' nonlinearity and compounding effects. To bridge the gap, we directly optimize orthogonal projection matrices for KV cache compression in LLMs with a distillation objective using an elaborate Matryoshka training strategy. After training, we show that adaptive compression rates for different layers and heads ensure optimal performance compared to uniform compression rates across all layers and heads in LLMs. Experimental results demonstrate significant performance gains and flexibility in achieving desired compression rates compared to traditional PCA projection.

ACKNOWLEDGEMENTS

This work was supported by NSF of China (Nos. 92470118, 62306176), Natural Science Foundation of Shanghai (No. 23ZR1428700), and CCF-Zhipu Large Model Innovation Fund (No. CCF-Zhipu202412).

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

## A    ERROR ANALYSIS

Consider in a decoder layer with key and value has same head dimension: $d = d_{k/v}$, our orthogonal projection $U^K, U^V \in \mathbb{R}^{d \times d}$ , we donate the first $r$ columns of orthogonal projection $U$ as $U_r$ and the rest as $U_{:,r:d}$, the error can be computed at a cache budget $r/d$ as:

$$
\begin{aligned}
\mathcal{L}(r) &= \left\| \text{Attention}(Q, K, V)W^O - \text{Attention}(\tilde{Q}, \tilde{K}, \tilde{V})W^{OV} \right\|_F^2 \\
&= \left\| \text{Softmax}\left(\frac{QK^\top}{\sqrt{d}}\right)VW^O - \text{Softmax}\left(\frac{\tilde{Q}\tilde{K}^\top}{\sqrt{d}}\right)\tilde{V}U_r^{V\top}W^O \right\|_F^2 \\
&= \left\| \text{Softmax}\left(\frac{QK^\top}{\sqrt{d}}\right)VU^VU^{V\top}W^O - \text{Softmax}\left(\frac{\tilde{Q}\tilde{K}^\top}{\sqrt{d}}\right)VU_r^VU_r^{V\top}W^O \right\|_F^2 \\
&= \left\| \text{Softmax}\left(\frac{QK^\top}{\sqrt{d}}\right)V\left(U_{:,r:d}^VU_{:,r:d}^{V\top} + U_r^VU_r^{V\top}\right)W^O - \text{Softmax}\left(\frac{\tilde{Q}\tilde{K}^\top}{\sqrt{d}}\right)VU_r^VU_r^{V\top}W^O \right\|_F^2 \\
&= \left\| \left( \text{Softmax}\left(\frac{QK^\top}{\sqrt{d}}\right) - \text{Softmax}\left(\frac{\tilde{Q}\tilde{K}^\top}{\sqrt{d}}\right)VU_r^VU_r^{V\top} + \text{Softmax}\left(\frac{QK^\top}{\sqrt{d}}\right)VU_{:,r:d}^VU_{:,r:d}^{V\top} \right) \right\|_F^2
\end{aligned}
\tag{2}
$$

Consider the original parameter $W_O$. By donating $\mathcal{L}_{QK} = \text{Softmax}\left(\frac{QK^\top}{\sqrt{d}}\right) - \text{Softmax}\left(\frac{\tilde{Q}\tilde{K}^\top}{\sqrt{d}}\right)$ and $A = \text{Softmax}\left(\frac{QK^\top}{\sqrt{d}}\right)$ is a constant, we just need to minimize:

$$\mathcal{L}\left(r\right) = \left\| \left(\mathcal{L}_{QK} V U_r^V U_r^{V\top} + A V U_{:,r:d}^V U_{:,r:d}^{V\top}\right) \right\|_F^2$$
$$= \left\| \mathcal{L}_{QK} + \left(A - \mathcal{L}_{QK}\right) V U_{:,r:d}^V U_{:,r:d}^{V\top} \right\|_F^2$$

(3)

While PCA on value states minimizes $\left\| V U_{:,r:d}^V U_{:,r:d}^{V\top} \right\|_F^2$ and PCA on query and key states minimizes $\mathcal{L}_{QK} = \text{Softmax}\left(\frac{QK^\top}{\sqrt{d}}\right) - \text{Softmax}\left(\frac{\tilde{Q}\tilde{K}^\top}{\sqrt{d}}\right)$, these optimizations do not necessarily guarantee the minimization of the global error $\mathcal{L}$, showing the PCA projection is suboptimal and has the room to be optimized to make the global error minimized.

The LLM itself has numerous layers, and each layer is nonlinear. Strictly speaking, the error is the output of the last layer of the model after low-rank projection and that of the original model's last layer. Here, we only conduct an intuitive analysis of a certain layer. The optimal solution of this optimization problem is complex and difficult to solve mathematically. So, we make these orthogonal matrices trainable to get optimal results.

Theoretically, the optimal solution also changes with the variation of the input data distribution. It is difficult for us to model the distribution of all corpora in the world. Therefore, to minimize the error of the model after KV cache compression on most tasks as much as possible, we consider using a data-driven approach for optimization to be a reasonable method.

To minimize $\mathcal{L}\left(r\right) = \left\| \text{Attention}(Q, K, V)W^O - \text{Attention}(\tilde{Q}, \tilde{K}, \tilde{V})W^{OV} \right\|_F^2$, we use KL-Divergence as a proxy loss to let the distributions of the two models' outputs close to each other. As we have discussed in Section 3.2, to recover the original $K$ and $V$ from the reduced $K'$ and $V'$ when using full-rank, the orthogonality of $U$ should be guaranteed. Thus, our optimization objective can be derived as:

$$U^* = \arg\min_U \sum_{r \in \mathcal{M}} D_{\text{KL}}\left(p\left(\cdot|\boldsymbol{x}\right) \| p'\left(\cdot|\boldsymbol{x}; U_r^K, U_r^V\right)\right)$$
$$\text{s.t. } D_{\text{KL}}\left(p\left(\cdot|\boldsymbol{x}\right) \| p'\left(\cdot|\boldsymbol{x}; U_d^K, U_d^V\right)\right) = 0$$

(4)

where we abuse $p'$ to refer to the LLM equipped with low-rank projection matrices, and $\mathcal{M}$ is our predefined schedule.

Our orthogonal constraints $UU^\top = I$ on $U$ can guarantee $D_{\text{KL}}\left(p\left(\cdot|\boldsymbol{x}\right) \| p'\left(\cdot|\boldsymbol{x}; U_d^K, U_d^V\right)\right) = 0$. It is worth noticing that if we only use $r < d$ columns to forward for KV cache compression, the rest $d - r$ columns, i.e. $U_{:,r:d}$ will not be updated. Thus, although experiments in Appendix G demonstrate that our method is not sensitive to a predefined schedule, we point out that $d \in \mathcal{M}$ is a must to guarantee all parameters of $U$ to be trained.

## B  WEIGHT MERGING METHOD

Given that both $W^Q$ and $W^K$, as well as our orthogonal projection, operate on hidden states, consolidating parameters evidently reduces computational time. However, many LLMs utilize RoPE Su et al. (2023), introducing a relative position embedding between $W^Q$ and $W^K$, which complicates integrating the parameters with our unitary transform. This issue has been addressed in prior works Saxena et al. (2024); Yu et al. (2024). The approach in Saxena et al. (2024) involves maintaining the merged parameters and transforming the compressed dimension cache back to its original dimensions for reapplication of RoPE. This does not reduce peak memory usage for attention and necessitates RoPE for all past tokens. Alternatively, (Yu et al., 2024) compresses the key states post-RoPE, which prohibits the merging of $W^{Q/K}$ and $U^K$. However, as only a single new token requires orthogonal transformation and dimensionality reduction during inference, the time increase is merely slight as shown in (Yu et al., 2024). Consequently, our treatment of RoPE in the present study is influenced by (Yu et al., 2024)'s methodology. The integration of the weight parameters of

$W^O$ and $U^{V\top}$, given RoPE has no impact on value states, the details of our weight merging methods can be formulated as follows and in Figure 5

$$
\begin{aligned}
\mathrm{MSA}(X) &= \mathrm{Concat}(\mathrm{head}_1, \mathrm{head}_2, \ldots, \mathrm{head}_H)W^O \\
&= \mathrm{Concat}(A_1V_1, A_2V_2, \cdots, A_hV_H)W^O \\
&= \mathrm{Concat}(A_1V_1U_1^V U_1^{V\top}, A_2V_2U_2^V U_2^{V\top}, \cdots, A_hV_hU_H^V U_H^{V\top})W^O \\
&= \mathrm{Concat}(A_1V_1U_1^V, A_2V_2U_2^V, \cdots, A_HV_HU_H^V)\left(\tilde{U}^V W^O\right) \\
&= \mathrm{Concat}(A_1\tilde{V}_1, A_2\tilde{V}_2, \cdots, A_H\tilde{V}_H)W^{OV}
\end{aligned}
$$
$$
\text{where} \quad A_i = \mathrm{Softmax}\left(\frac{Q_iK_i^\top}{\sqrt{d_k}}\right) \text{ is the attention weights of a given head in each layer}
$$

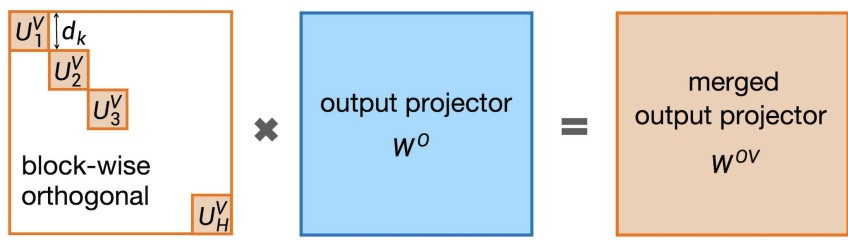

Figure 5: After obtaining an orthogonal matrix through training, we merge the parameters in this way, reducing the number of matrix multiplications required during inference without incurring any inference time overhead. Truncation can be achieved simply by removing the columns corresponding to $W_{OV}$, thereby reducing peak memory consumption.

## C  TWO STAGE SFT

In this section, we provide a detailed discussion on our observations regarding fine-tuning with LoRA and the orthogonal matrix. We elaborate on the issues stemming from calculating covariance on a limited sample subset and performing spectral decomposition, which may lead to suboptimal parameters. We hypothesize that larger gradients during training can arise from task-specific distributions, such as in GSM8K, affecting the alignment of LoRA weights with the base model.

To mitigate these issues, our two-phase training approach involves initially training only the LoRA weights to ensure adequate adaptation to downstream tasks. In the second phase, we introduce simultaneous training of the unitary transformation matrix and the LoRA weights, focusing on maintaining performance while compressing the cache effectively. We also explore the impact of using separate learning rates for the LoRA and orthogonal matrix parameters to further investigate these phenomena. Extensive experimental results are provided to support our findings.

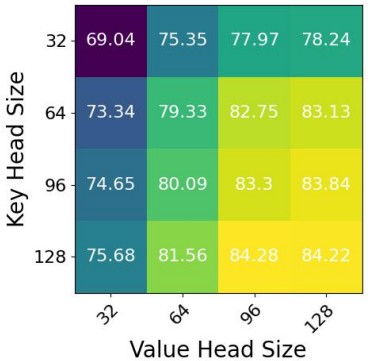

Figure 6: Two-phase SFT on PIQA.

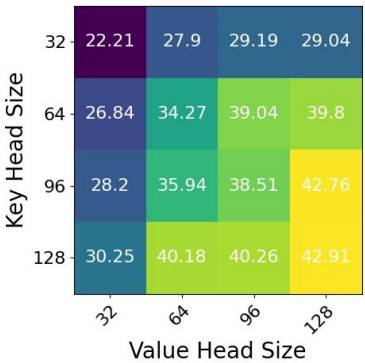

Figure 7: Two-phase SFT on GSM8K.

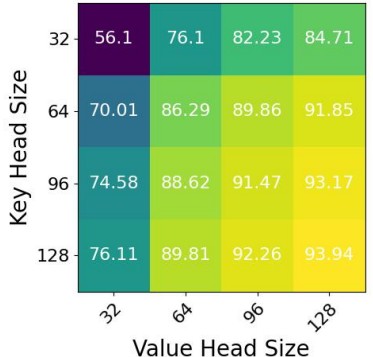

Figure 8: Two-phase SFT on HellaSwag.

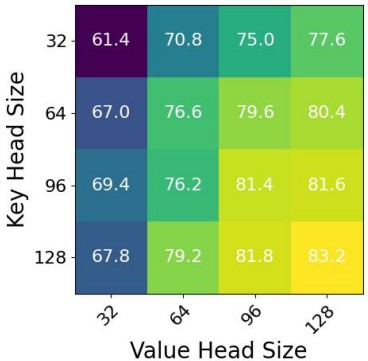

Figure 9: Two-phase SFT on OBQA.

# D ABLATION STUDY ON GREEDY SEARCH FOR ADAPTIVE COMPRESSION LEVELS

We present some experimental results using a uniform compression rate across all heads after CPT and SFT in our MatryoshkaKV LLM. The results are displayed in 4.

Table 4: Accuracy of our distilled MatryoshkaKV Projections after CPT on six benchmarks w/o greedy search for adaptive compression levels.

| Model | Budget | Method | HLSG | ARC-C | ARC-E | PIQA | WG | CSQA | Avg. |
|---|---|---|---|---|---|---|---|---|---|
| LLaMA2 7B-base | 100.0% | baseline | 74.00 | 35.93 | 50.97 | 78.50 | 61.64 | 65.93 | 61.16 |
| | | PCA | 72.04 | 36.95 | 52.38 | 76.66 | 61.72 | 67.24 | 61.17 |
| | | MKV | 72.05 | 37.29 | 52.38 | 76.66 | 61.72 | 67.32 | 61.24 |
| | 87.5% | PCA | 30.28 | 23.73 | 30.34 | 58.05 | 51.30 | 22.60 | 36.05 |
| | | MKV | 72.22 | 35.93 | 52.20 | 76.28 | 62.12 | 65.27 | 60.67 |
| | 75.0% | PCA | 25.47 | 27.80 | 27.51 | 52.67 | 49.72 | 20.56 | 33.96 |
| | | MKV | 70.98 | 34.58 | 55.20 | 76.77 | 61.56 | 63.64 | 60.46 |
| | 62.5% | PCA | 24.22 | 28.81 | 27.51 | 51.58 | 50.28 | 21.29 | 33.95 |
| | | MKV | 69.22 | 37.29 | 55.73 | 75.22 | 59.35 | 64.21 | 60.17 |
| | 50.0% | PCA | 24.04 | 28.47 | 25.22 | 52.29 | 50.67 | 20.72 | 33.57 |
| | | MKV | 66.62 | 34.24 | 52.91 | 75.46 | 58.41 | 62.00 | 58.27 |
| | 37.5% | PCA | 24.08 | 28.47 | 25.40 | 50.76 | 49.49 | 18.35 | 32.76 |
| | | MKV | 62.38 | 32.20 | 50.26 | 73.34 | 56.67 | 55.28 | 55.02 |
| | 25.0% | PCA | 23.98 | 29.49 | 26.28 | 51.20 | 50.36 | 16.22 | 32.92 |
| | | MKV | 51.91 | 27.46 | 44.44 | 69.64 | 54.54 | 44.39 | 48.73 |
| Mistral-v0.3 7B-base | 100.0% | baseline | 75.50 | 42.03 | 63.14 | 80.25 | 65.43 | 70.68 | 66.17 |
| | | PCA | 75.46 | 42.03 | 62.96 | 80.25 | 65.35 | 70.27 | 66.05 |
| | | MKV | 75.44 | 42.03 | 62.96 | 80.25 | 65.51 | 70.27 | 66.08 |
| | 87.5% | PCA | 37.09 | 22.03 | 34.57 | 59.85 | 53.67 | 33.99 | 40.20 |
| | | MKV | 77.01 | 42.37 | 62.43 | 80.09 | 65.51 | 70.52 | 66.32 |
| | 75.0% | PCA | 30.58 | 20.68 | 30.86 | 58.92 | 51.14 | 24.65 | 36.14 |
| | | MKV | 75.55 | 40.34 | 63.49 | 80.47 | 64.48 | 70.60 | 65.82 |
| | 62.5% | PCA | 28.91 | 21.69 | 26.46 | 56.58 | 51.14 | 21.70 | 34.41 |
| | | MKV | 73.95 | 38.98 | 62.61 | 79.22 | 64.40 | 68.39 | 64.59 |
| | 50.0% | PCA | 27.40 | 23.73 | 26.28 | 55.01 | 50.43 | 22.77 | 34.27 |
| | | MKV | 71.65 | 36.95 | 60.85 | 78.40 | 62.19 | 66.91 | 62.83 |
| | 37.5% | PCA | 25.77 | 21.69 | 24.34 | 53.70 | 49.57 | 21.46 | 32.76 |
| | | MKV | 68.63 | 33.56 | 56.26 | 77.48 | 59.83 | 62.16 | 59.65 |
| | 25.0% | PCA | 24.91 | 26.10 | 25.40 | 52.67 | 48.30 | 19.74 | 32.85 |
| | | MKV | 59.21 | 25.42 | 48.68 | 73.83 | 54.30 | 45.13 | 51.10 |

# E COMPATIBILITY WITH OTHER KV CACHE COMPRESSION TECHNIQUES

In this appendix, we present detailed results and analysis on the combination of MatryohskaKV with two orthogonal key-value (KV) cache compression techniques: $H_2O$ (Zhang et al., 2023b) and KIVI (Liu et al., 2023). Both combinations are evaluated on the datasets mentioned in Section 5.1, and their performance is summarized in Table 5 and Table 6, respectively.

Table 5: Results of Combination of Distilled MatryoshkaKV Projections and $H_2O$ across Seven Benchmarks. We use uniform compression levels for inference here for simplicity. The first and second columns indicate the individual compression rates along two axes. If $H_2O$ uses 20% cache on the sequence length axis and MatryoshkaKV uses 50% cache on the feature dimension axis, the overall cache utilization is 10%.

| $H_2O$ | MKV | LongBench | HLSG | ARC-C | ARC-E | PIQA | WG | CSQA | Avg. |
|---|---|---|---|---|---|---|---|---|---|
| 100 % | 100% | 4.17 | 72.05 | 37.29 | 52.38 | 76.66 | 61.72 | 67.32 | 61.24 |
| | 87.5% | 4.44 | 72.22 | 35.93 | 52.20 | 76.28 | 62.12 | 65.27 | 60.67 |
| | 75.0% | 4.57 | 70.98 | 34.58 | 55.20 | 76.77 | 61.56 | 63.64 | 60.46 |
| | 62.5% | 4.70 | 69.22 | 37.29 | 55.73 | 75.22 | 59.35 | 64.21 | 60.17 |
| | 50.0% | 4.93 | 66.62 | 34.24 | 52.91 | 75.46 | 58.41 | 62.00 | 58.27 |
| | 37.5% | 5.47 | 62.38 | 32.20 | 50.26 | 73.34 | 56.67 | 55.28 | 55.02 |
| | 25.0% | 7.66 | 51.91 | 27.46 | 44.44 | 69.64 | 54.54 | 44.39 | 48.73 |
| 75 % | 100% | 4.18 | 70.71 | 36.61 | 52.38 | 76.55 | 60.54 | 66.50 | 60.55 |
| | 87.5% | 4.44 | 71.42 | 35.25 | 53.09 | 76.33 | 59.91 | 64.62 | 60.74 |
| | 75.0% | 4.57 | 70.31 | 34.34 | 54.14 | 76.39 | 59.27 | 62.90 | 59.94 |
| | 62.5% | 4.70 | 68.47 | 36.27 | 54.32 | 75.41 | 58.48 | 63.96 | 59.89 |
| | 50.0% | 4.94 | 66.00 | 32.54 | 51.50 | 75.63 | 57.30 | 61.43 | 57.46 |
| | 37.5% | 5.47 | 61.50 | 32.88 | 49.21 | 73.01 | 55.09 | 55.12 | 54.63 |
| | 25.0% | 7.67 | 51.32 | 27.80 | 44.09 | 69.37 | 53.59 | 44.55 | 48.47 |
| 50 % | 100% | 4.20 | 68.72 | 33.22 | 52.20 | 76.12 | 56.67 | 64.78 | 58.62 |
| | 87.5% | 4.46 | 67.89 | 34.58 | 51.85 | 76.28 | 55.88 | 62.00 | 58.13 |
| | 75.0% | 4.59 | 66.01 | 35.59 | 53.79 | 75.41 | 54.54 | 62.00 | 58.05 |
| | 62.5% | 4.73 | 63.59 | 34.92 | 51.32 | 75.68 | 55.25 | 60.52 | 57.04 |
| | 50.0% | 4.96 | 61.33 | 36.10 | 50.74 | 73.67 | 55.57 | 57.67 | 55.85 |
| | 37.5% | 5.50 | 59.26 | 29.83 | 49.91 | 73.61 | 53.04 | 54.14 | 53.29 |
| | 25.0% | 7.71 | 49.44 | 26.44 | 41.80 | 68.72 | 52.96 | 43.24 | 46.94 |
| 20 % | 100% | 4.40 | 61.55 | 25.76 | 41.27 | 73.29 | 53.28 | 47.01 | 49.98 |
| | 87.5% | 4.65 | 61.36 | 30.51 | 39.86 | 73.72 | 52.09 | 49.06 | 50.94 |
| | 75.0% | 4.79 | 60.29 | 28.47 | 38.62 | 72.75 | 53.12 | 50.45 | 50.62 |
| | 62.5% | 4.93 | 58.77 | 26.78 | 39.86 | 70.84 | 52.72 | 49.30 | 50.58 |
| | 50.0% | 5.19 | 56.39 | 26.78 | 38.10 | 71.22 | 51.62 | 49.16 | 49.66 |
| | 37.5% | 5.74 | 52.12 | 23.39 | 34.22 | 68.50 | 52.17 | 41.44 | 44.82 |
| | 25.0% | 8.01 | 43.22 | 21.02 | 31.92 | 63.93 | 51.38 | 33.09 | 40.75 |

## F    COMPARISONS WITH MORE BASELINES

We introduce an additional baseline, ASVD(Yuan et al., 2024), which has been developed to address the low-rank characteristics of LLM parameters. This approach performs simultaneous compression of both the KV cache and the model parameters, allowing for efficient utilization of memory resources. ASVD provides checkpoints for three specific cache budgets: 85%, 90%, and 95%. In our experiments, we compare our MatryoshkaKV against ASVD under these budgets to evaluate performance and efficiency. The results of these comparisons are detailed in Table 7, where we present the performance metrics for our method alongside those obtained using ASVD.

We evaluate the inference speed of our LLM equipped with MatryoshkaKV and compare it to the LLaMA2-7B-base model. The results are displayed in Table 8. Specifically, these evaluations were conducted during the inference process with a batch size of 32. Our current implementation consumes a slightly faster time than the baseline full-KV model. This is because we have not performed system-level optimizations for memory copy and sparse computations involved in our KV mechanism.

Table 6: Results of Combination of Distilled MatryoshkaKV Projections and KIVI (2bit KV cache quantization) on Six Benchmarks. We use uniform compression levels for inference here for simplicity.

| Model | Budget | Method | HLSG | ARC-C | ARC-E | PIQA | WG | CSQA | Avg. |
|-------|--------|--------|------|-------|-------|------|-----|------|------|
| LLaMA2 7B-base | 100.0% | MKV | 70.89 | 36.95 | 53.26 | 76.39 | 61.56 | 67.08 | 60.98 |
| | | MKV+KIVI | 69.76 | 35.93 | 51.98 | 76.55 | 61.48 | 66.26 | 60.49 |
| | 87.5% | MKV | 70.87 | 36.95 | 51.15 | 76.17 | 61.80 | 64.95 | 60.47 |
| | | MKV+KIVI | 70.45 | 36.61 | 50.79 | 76.44 | 61.01 | 64.13 | 59.91 |
| | 75.0% | MKV | 69.30 | 33.90 | 54.67 | 75.90 | 61.09 | 63.23 | 60.08 |
| | | MKV+KIVI | 68.62 | 32.20 | 54.85 | 76.06 | 60.30 | 63.23 | 59.68 |
| | 62.5% | MKV | 67.25 | 36.27 | 53.62 | 75.52 | 59.27 | 64.46 | 59.39 |
| | | MKV+KIVI | 66.56 | 35.25 | 51.68 | 75.41 | 59.43 | 61.59 | 58.33 |
| | 50.0% | MKV | 65.08 | 33.56 | 52.03 | 74.81 | 57.54 | 60.36 | 56.98 |
| | | MKV+KIVI | 63.25 | 32.54 | 51.15 | 74.43 | 57.38 | 59.46 | 56.35 |
| | 37.5% | MKV | 61.02 | 29.83 | 49.21 | 73.45 | 55.64 | 55.36 | 54.09 |
| | | MKV+KIVI | 57.11 | 28.81 | 48.85 | 71.71 | 55.64 | 50.37 | 52.08 |
| | 25.0% | MKV | 50.61 | 25.76 | 45.33 | 69.64 | 54.30 | 43.90 | 47.96 |
| | | MKV+KIVI | 48.12 | 27.80 | 42.86 | 67.85 | 53.59 | 40.54 | 46.78 |

Table 7: Comparison between our MatryoshkaKV and baseline ASVD. We use uniform compression levels for inference here for simplicity.

| Model | Budget | Method | HLSG | ARC-C | ARC-E | PIQA | WG | CSQA | Avg. |
|-------|--------|--------|------|-------|-------|------|-----|------|------|
| LLaMA2 7B-base | 100.0% | baseline | 74.00 | 35.93 | 50.97 | 78.50 | 61.64 | 65.93 | 61.16 |
| | 95% | ASVD | 71.12 | 36.95 | 52.20 | 76.28 | 62.35 | 66.67 | 60.92 |
| | | MKV | 72.59 | 36.27 | 53.09 | 76.44 | 62.43 | 66.75 | 61.25 |
| | 90% | ASVD | 70.45 | 34.92 | 52.03 | 75.63 | 61.72 | 64.70 | 60.06 |
| | | MKV | 72.30 | 36.61 | 54.50 | 76.50 | 62.90 | 65.93 | 62.03 |
| | 85% | ASVD | 67.23 | 35.93 | 50.26 | 74.86 | 60.38 | 62.16 | 59.29 |
| | | MKV | 72.33 | 35.93 | 53.26 | 76.33 | 61.80 | 64.78 | 61.13 |

# G   EXPERIMENTS ON VARIOUS HYPER-PARAMETERS

In Section 5.1, during the training process, we initially predefine the schedule set as $\left\{\frac{i}{8}d\right\}_{i=1}^{8}$. Subsequently, we modify the schedule set to $\left\{\frac{i}{4}d\right\}_{i=1}^{4}$ while keeping other hyper-parameters unchanged. Then, we evaluate the accuracy using the same benchmarks. The results are listed in Table 9:

Table 8: Tokens per second at different percentages.

|  | LLaMA2 | 100% | 87.5% | 75% | 62.5% | 50% | 37.5% | 25% |
|---|---|---|---|---|---|---|---|---|
| Tokens per second | 33.65 | 34.12 | 34.08 | 34.90 | 35.27 | 36.42 | 36.75 | 37.22 |

Table 9: Accuracy of our MatryoshkaKV after CPT on six benchmarks. We use uniform compression levels for inference here for simplicity. Different hyper-parameters are compared. In the table we donate the schedule $\{\frac{i}{8}d\}_{i=1}^{8}$ as $\mathcal{M}_2$, and the schedule $\{\frac{i}{4}d\}_{i=1}^{4}$ as $\mathcal{M}_1$. We use uniform compression levels for inference here for simplicity.

| Model | Budget | Method | HLSG | ARC-C | ARC-E | PIQA | WG | CSQA | Avg. |
|---|---|---|---|---|---|---|---|---|---|
| LLaMA2 7B-base | 100.0% | $\mathcal{M}_1$ | 72.03 | 36.61 | 52.56 | 76.71 | 61.64 | 67.16 | 62.07 |
|  |  | $\mathcal{M}_2$ | 72.05 | 37.29 | 52.38 | 76.66 | 61.72 | 67.32 | 61.24 |
|  | 87.5% | $\mathcal{M}_1$ | 72.03 | 37.29 | 53.09 | 76.28 | 62.75 | 65.77 | 62.18 |
|  |  | $\mathcal{M}_2$ | 72.22 | 35.93 | 52.20 | 76.28 | 62.12 | 65.27 | 60.67 |
|  | 75.0% | $\mathcal{M}_1$ | 70.79 | 34.92 | 53.62 | 76.88 | 60.54 | 65.03 | 61.31 |
|  |  | $\mathcal{M}_2$ | 70.98 | 34.58 | 55.20 | 76.77 | 61.56 | 63.64 | 60.46 |
|  | 62.5% | $\mathcal{M}_1$ | 69.03 | 32.88 | 52.91 | 74.86 | 59.19 | 64.54 | 59.69 |
|  |  | $\mathcal{M}_2$ | 69.22 | 37.29 | 55.73 | 75.22 | 59.35 | 64.21 | 60.17 |
|  | 50.0% | $\mathcal{M}_1$ | 66.34 | 32.88 | 53.09 | 74.97 | 58.25 | 62.49 | 58.59 |
|  |  | $\mathcal{M}_2$ | 66.62 | 34.24 | 52.91 | 75.46 | 58.41 | 62.00 | 58.27 |
|  | 37.5% | $\mathcal{M}_1$ | 61.55 | 31.19 | 49.91 | 73.83 | 56.27 | 52.09 | 53.78 |
|  |  | $\mathcal{M}_2$ | 62.38 | 32.20 | 50.26 | 73.34 | 56.67 | 55.28 | 55.02 |
|  | 25.0% | $\mathcal{M}_1$ | 50.91 | 26.10 | 44.97 | 68.39 | 52.72 | 38.33 | 46.38 |
|  |  | $\mathcal{M}_2$ | 51.91 | 27.46 | 44.44 | 69.64 | 54.54 | 44.39 | 48.73 |

The final results of our MatryoshkaKV are not very sensitive to the schedule choice.

