# OpenReview forum: "MatryoshkaKV: Adaptive KV Compression via Trainable Orthogonal Projection"
_ICLR.cc/2025/Conference — ICLR 2025 Poster_

### Official Review · Reviewer_9USD · 2024-11-02

**Soundness:** 2
**Presentation:** 3
**Contribution:** 3
**Rating:** 6
**Confidence:** 4

**Summary:**

This work proposes a KV cache compression technique for efficient LLM inference through low-rank projections along the feature dimension.  Using principal component analysis (PCA) over key and value matrices to obtain low-rank counterparts leads to performance loss, especially at high compression ratios. To tackle this, the authors propose to tune the projection matrices for each layer and attention head with a distillation loss. Results are presented for Llama-2 (7b) and Mistral-v0.3 (7b) models.

**Strengths:**

1. This work targets KV cache compression, a crucial problem for efficient LLM inference at large sequence lengths. Results show extensive improvements over vanilla PCA for a variety of downstream tasks.

**Weaknesses:**

1. The training to obtain orthogonal projection matrices involves a KL divergence loss, which ensures that the KV compressed model performance stays close to the original model. However, this makes the strategy task-dependent by using a form of calibration dataset for the downstream task itself, leading to the necessity of training every time one needs a compressed KV cache for performing inference on certain task(s). Additionally, there is no clarification on the calibration dataset used for continual pretraining experiments in Section 5.1.
2. For results in Section 5.2, the authors propose a 2-stage training pipeline: 1. LoRA (standard fine-tuning) 2. Updating projection matrices and LoRA parameters jointly. While the first stage is generally employed to improve the performance of LLMs on downstream tasks, the second stage is the associated overhead with this form of KV cache compression. Additionally, the joint update in this stage implies the need to compress the KV cache specifically for each downstream task. It would be ideal to have a task-agnostic KV cache compression scheme.
3. Comparisons with baselines are missing and/or somewhat ambiguous. Is the PCA baseline in Table 1 the same as Eigen Attention [1]? Another missing potential baseline is ASVD [2], which also involves training-free low-rank projection to reduce the KV cache footprint. The authors clarify that a variety of works compress the KV cache by targeting the sequence length or channel dimension, but don't demonstrate the possibility of using their approach concurrently with such techniques [3,4].
4. Results on the more recent family of Llama models (Llama 3/3.1) are missing but crucial to establish the effectiveness of this approach.

[1] Saxena et al., "Eigen Attention: Attention in Low-Rank Space for KV Cache Compression.", ArXiv 2024.

[2] Yuan et al., "ASVD: Activation-aware Singular Value Decomposition for Compressing Large Language Models", ArXiv 2024.

[3] Zhang et al., "H2O: Heavy-Hitter Oracle for Efficient Generative Inference of Large Language Models", NeurIPS 2023.

[4] Liu et al., "KIVI: A Tuning-Free Asymmetric 2bit Quantization for KV Cache", ICML 2024.

**Questions:**

1. Can the authors clarify how they achieve performance benefits with heterogeneous ranks across the head dimension? Different dimensions across the heads may require some form of padding before concatenating them during actual inference, so it would be great to see some hardware performance numbers as well.
2. Which calibration dataset was used for the results presented in Table 1?
3. For the SFT setup described in Section 5.2, it would be interesting to see if the second stage of training can just be limited to projection tuning instead of the proposed joint tuning with LoRA parameters as well.

---

> ### Author Response · Authors · 2024-11-20
> **Response to Reviewer 9USD (Part 1/2)**
>
> Thank you for your careful reading and many pertinent suggestions.
> We are also very grateful for your advice on supplementary experiments.
> We have revised the unclear parts in the paper as quickly as possible and presented the results according to your experimental suggestions.
>
> **W1 (Q2):** Which calibration dataset was used for the results presented in Table 1? The necessity of training every time?
>
> Firstly, we would like to clarify that in the continual pretraining scenario, **our approach encompasses two datasets: a training set, Redpajama, which is task-agnostic and utilized for training our orthogonal matrices via single-time continue pre-training (CPT), and a calibration set which is task-dependent and employed solely for the greedy search to attain adaptive compression rates (training-free) before inference on the specific task**.
>
> Secondly, the calibration set is randomly sampled from the downstream dataset and composed of merely 5-10 samples. We will have a more clear description in the revised version.
>
> **Q1:** Can the authors clarify how they achieve performance benefits with heterogeneous ranks across the head dimension?
>
> Indeed, what we have done based on padding currently results in an actual time that is even slightly worse than that of the uniform approach.
> However, this can be resolved through system-level optimizations (e.g., flattening the caches of different heads for storage, see [AdaKV](https://github.com/FFY0/AdaKV)), which is not the main technical focus of this paper.
> The upper bound of the theoretical acceleration of our approach exists.
>
> On the other hand, in practice, our uniform strategy is also quite powerful.
> It can also achieve an average accuracy of 55.02 on 6 benchmarks with a 37.5% cache budget, which is 89.96% of the baseline.
> The uniform strategy doesn't require those system implementations and takes about the same amount of time as the original attention in real-world scenarios as listed below.
>
> The inference speed of our MatryoshkaKV under the uniform compression rate during the inference process with a batch size of 32.
> |      | LLaMA2-7B-base | 100% | 87.5% | 75% | 62.5% | 50% | 37.5% | 25% |
> |------|------|------|------|------|------|------|------|------|
> | Tokens per second | 33.65 | 34.12 | 34.08 | 34.90 | 35.27 | 36.42 | 36.75 | 37.22 |
>
> We clarify our paper mainly focuses on reducing storage and memory transfer costs instead of runtime, and we have proven that our approach indeed consumes significantly less storage cost than the baseline.
> We will make these points clear in the revised revision.
>
>
> [1]: Roberts et al., "TensorNetwork: A Library for Physics and Machine Learning", arXiv 2019.
>
> [2]: Saxena et al., "Eigen Attention: Attention in Low-Rank Space for KV Cache Compression.", ArXiv 2024.
>
> [3]: Yuan et al., "ASVD: Activation-aware Singular Value Decomposition for Compressing Large Language Models", ArXiv 2024
>
> [4]: Liu, A., Liu, J., Pan, Z., He, Y., Haffari, G., & Zhuang, B. "MiniCache: KV Cache Compression in Depth Dimension for Large Language Models", NeurIPS 2024.

---

> ### Author Response · Authors · 2024-11-20
> **Response to Reviewer 9USD (Part 2/2)**
>
> **W2 (Q3):** It would be ideal to have a task-agnostic KV cache compression scheme. If the second stage of training can just be limited to projection tuning？
>
> We clarify our experiments on continual pretraining are what you desired. We add the SFT experiments to only prove that our method can effectively handle certain specific or more challenging tasks (such as GSM8K).
> Generally, LLMs also require task-specific SFT for handling such tasks.
>
> Moreover, the advantage of our 2-stage training pipeline is that, **compared to CPT, it has lower costs and requires less time**.
> For example, for our SFT on GSM8K, the first stage only consumes 0.5 A100(40G) GPU $\times$ hour, and the second stage only consumes 1.5 A100(40G) GPU $\times$ hour.
> This is much less than the cost of the CPT scenario, which consumes 30 A100 GPU $\times$ hour.
> Despite the lower cost, satisfactory performance can be achieved on more difficult tasks (such as GSM8K).
> Compared with the normally LoRA-finetuned Llama2-7b (Lora rank is set as 8), we can achieve approximately 90% accuracy with only 50% of the KV cache.
>
> We have attempted to tune merely orthogonal projections, but we find that this would significantly affect the convergence of the model.
> We believe that for some difficult downstream tasks such as GSM8K, perhaps the model requires a higher degree of freedom to fit this data distribution.
>
> **W3:** Baselines & Using the approach concurrently with other techniques.
>
> Thanks for your advice on supplementing baselines.
>
> We first clarify that our PCA baseline is roughly identical to Eigen-Attention [2], and the reproduction details are described in Section 3.2).
> The results of the PCA baseline in our paper are similar to those in Eigen-Attention [2] (see Table 2 in [2] and Table 1 in ours).
> We will make it clear in the upcoming revisions.
>
> Besides, **we have incorporated ASVD [3] as our baseline, the results are supplemented in Table 3 in General Reply**.
>
> As ASVD only furnishes checkpoints for three cache budgets, namely 85%, 90%, and 95%, we have thus compared our method with ASVD within these three specific budget scenarios.
> The results exhibit that our MatyoshkaKV performs better than ASVD at all three cache budget levels.
>
>
> **In Section 5, we apply MatryoshkaKV to Mistral-v0.3-7B-base, and it evidently demonstrates enhanced compression capabilities when used together with Group Query Attention (GQA).
> Moreover, we also combine our methods with H2O(token eviction and merging) and KIVI (a KV quantization technique)**
>
>
>
> For the combination with H2O, our results are supplemented in **Table 1 in General Reply**. According to the results, when MatryoshkaKV and H2O are used concurrently, the perplexity on long contexts only increases by 1.02 at a 10% KV cache budget.
> Additionally, if we compress by 50% on both the sequence length and feature dimension axes (with an actual cache usage rate of 25%), we can achieve an average accuracy of 55.85 on 6 benchmarks, which is 91.32% of the baseline.
> This is much better than the effect of only using 25% separately on these two individual axes.
>
> For the combination with KIVI, our results are supplemented in **Table 2 in General Reply**. The combination of MatryoshkaKV and KIVI doesn't lead to a significant decline in accuracy on six benchmarks mentioned in Section 5, showing that MatryoshkaKV can be integrated with KV quantization to achieve a higher compression ratio.
>
> **W4:** Results on the more recent family of Llama models (Llama 3/3.1) are missing.
>
> Thank you for the comment. First of all, we would like to clarify that we have covered multiple architectures, including LLaMA2-7B-base and Mistral-7B-v0.3-base.
> The difference between LlaMA3 and LLaMA2 is not significant, which mainly lies in whether GQA is used.
> We've already demonstrated that our method can be compatible with GQA by applying our MatryoshkaKV to Mistral-7B-v0.3-base.
> Therefore, it can be presumed that our method basically works for LLaMA3 as well.
> Additionally, related works such as MiniCache [4] have also conducted experiments on LlaMA2 and Mistral.
> We will attempt to include the results of Llama3 in the final version.
>
> [1]: Roberts et al., "TensorNetwork: A Library for Physics and Machine Learning", arXiv 2019
>
>
> [2]: Saxena et al., "Eigen Attention: Attention in Low-Rank Space for KV Cache Compression.", ArXiv 2024.
>
> [3]: Yuan et al., "ASVD: Activation-aware Singular Value Decomposition for Compressing Large Language Models", ArXiv 2024
>
> [4]: Liu, A., Liu, J., Pan, Z., He, Y., Haffari, G., & Zhuang, B. "MiniCache: KV Cache Compression in Depth Dimension for Large Language Models", NeurIPS 2024.

---

> ### Comment · Reviewer_9USD · 2024-11-24
> **Updated Rating**
>
> The authors have resolved most of my concerns, so I am happy to increase my score.
>
> I have one question related to the task-dependent greedy search: does it mean that whenever a model is used for inference on a new task, it needs to go through the greedy search for adaptive compression rates? Can the authors clarify the increase in inference latency/run-time due to this step?

---

> > ### Author Response · Authors · 2024-11-25
> > **Response to Reviewer 9USD**
> >
> > We group 8 task-dependent samples into a single batch and perform a unified forward pass to measure the deviation of the model's output from the original. A total of approximately 160 forward passes are required, taking around 3–4 minutes on a single 40GB A100 GPU.
> >
> > On one hand, we update the compression ratios of multiple heads in parallel (we updated the compression ratios of 32 heads in one go), which significantly reduces the time of greedy search while maintaining its effectiveness.
> >
> > On the other hand, since the adaptive rate for a specific task remains constant, these adaptive rates can be pre-calculated.
> > Thus, when testing on the same task again, the greedy search operation is no longer necessary.

---

> > ### Author Response · Authors · 2024-11-25
> > **Thanks for your thoughtful and comprehensive feedback**
> >
> > We sincerely thank you for taking the time to reassess our work. Your valuable suggestions have been instrumental in helping us improve and refine our manuscript.

---

### Official Review · Reviewer_76BC · 2024-11-04

**Soundness:** 3
**Presentation:** 2
**Contribution:** 3
**Rating:** 6
**Confidence:** 3

**Summary:**

The paper presents a novel method to efficiently compress Key-Value (KV) cache for large language models (LLMs) like LLaMA2-7B and Mistral-7B, which can be a bottleneck in terms of storage and memory transfer. The authors propose using low-rank orthogonal projection matrices, initialized with PCA and further fine-tuned with a distillation objective, to compress the feature dimension of the KV cache. The novel Matryoshka training strategy allows adaptive selection of compression levels for different layers and heads, balancing model performance and compression rate. Experiments demonstrate high data efficiency, with the method achieving over 90% performance while compressing the KV cache by 60% on average.

**Strengths:**

- The paper introduces a novel method to compress KV cache by focusing on the feature dimension. By employing low-rank projection matrices combined with orthogonality constraints, the authors efficiently reduce the KV cache size without requiring retraining from scratch, allowing the compression mechanism to be integrated directly into pre-trained models.

- The proposed training strategy to fine-tune orthogonal projection matrices effectively preserves model performance while allowing adaptive compression rates, providing a flexible approach to balance resource usage.

- The use of heterogeneous compression rates across different layers and heads is well-motivated and effectively demonstrated

**Weaknesses:**

- Although the paper briefly mentions other KV cache compression methods, such as sharing KV headers across layers and merging redundant tokens, it lacks a detailed comparison to highlight the advantages of the feature-dimension based compression. Including experimental comparisons or a more thorough discussion of the advantages and disadvantages of each approach would strengthen the contribution and clarify the unique benefits of the proposed method.

- The justification for using predefined schedules for Matryoshka strategy and the heterogeneous compression rates with greedy search could be made stronger with more theoretical backing or detailed analysis. For example, are the results sensitive to different schedules? Is the greedy search algo. deterministic and with the grantee to converge, and how's the scalability?

- The Figure 4 seems to indicate the Matryoshka strategy is much more important than greedy search (yellow and green lines are closed in right), and Orthogonal Constraint has less effect when Cache Utilization<0.5. More discussion and analysis on these findings are encouraged.

**Questions:**

see weakness

---

> ### Author Response · Authors · 2024-11-20
> **Response to Reviewer 76BC (Part 1/2)**
>
> We would like to express our sincere gratitude for your in-depth and insightful comments on our paper.
> Your feedback has provided us with valuable directions for improvement, and we truly appreciate the time and effort you have dedicated to reviewing our work.
>
> **W1:** It lacks a detailed comparison to highlight the advantages of the feature-dimension based compression
>
> Thanks for the question. We clarify that there are no specific advantages of feature-dimension based compression compared to works on other dimensions. Nevertheless, the exploration of compression in the feature dimension has been largely uncharted previously due to the inherent difficulties of compressing features of deep models.
>
> The enabling of the combination of feature-dimension compression with works on other dimensions is another important contribution of this work.
> We have demonstrated the ability to combine our MatryoshkaKV with the Group Query Attention (GQA) method in Section 5, by equipping Mistral-7V-v0.3-base with our trainable projections.
>
> Additionally, we combined our methods with H2O (token eviction and merging)[1] and KIVI (KV quantization)[2], and the results are presented in **Table 1 in General Reply** and **Table 2 in General Reply**.
>
> **W2:** The justification for using predefined schedules for the Matryoshka strategy and the heterogeneous compression rates with greedy search could be made stronger with more theoretical backing or detailed analysis.
>
> Thanks for the comment. We clarify that **our training approach is not highly sensitive to the selection of the schedule**. We have added experiments using other schedules in **Table 4 in General Reply**. It can be observed that the results do not change significantly.
>
> As for the greedy search for the adaptive compression levels algorithm, **it is only employed after the training process to obtain heterogeneous compression rates**. During the inference phase, this rate remains unchanged.
> Therefore, it has no connection with whether the training converges or not. During the training, we merely **randomly mask the feature dimension of each head**. We have found that normal convergence can be achieved in this way.
>
>
> [1]:  Zhang et al., "H2O: Heavy-Hitter Oracle for Efficient Generative Inference of Large Language Models", NeurIPS 2023.
>
> [2]: Liu et al., "KIVI: A Tuning-Free Asymmetric 2bit Quantization for KV Cache", ICML 2024.
>
> [3]: Aditya Kusupati, et al. Matryoshka representation learning. Advances in Neural Information Processing Systems, 2022.
>
> [4]: DeepSeek-AI at al., "DeepSeek-V2: A Strong, Economical, and Efficient Mixture-of-Experts Language Model", arXiv 2024.

---

> ### Author Response · Authors · 2024-11-20
> **Response to Reviewer 76BC (Part 2/2)**
>
> **W3 (part 1):** Figure 4 seems to indicate the Matryoshka strategy is much more important than the greedy search
>
> Thanks for your comments.
> Matryoshka strategy **employs randomly sampled compression ratios on each attention head during the training process**.
> It ensures the hierarchization of the orthogonal matrices and decouples the compression ratios of different heads during training, enabling users to arbitrarily choose KV cache compression ratios for tasks of different difficulties.
> As shown in Figure 4 right, without the Matryoshka strategy, our orthogonal projections fail to adapt and perform effectively at compression ratios that are not encountered during the training phase.
>
> Our greedy search algorithm is to set different ratios for different heads, which aligns with the observed phenomenon of anisotropy in different heads as reported in some related works we've discussed in Section 5.4.
>
> **It is noted that without the Matryoshka strategy, it is impossible for us to find the optimal ratio on each attention head through greedy search**. Therefore, the two need to be used in combination.
> On the one hand, each head may **only** perform well at the trained static compression ratio.
> On the other hand, the compression ratios between each head are also mutually coupled.
> This makes it impossible for us to use the greedy search algorithm to find the **independent** compression ratio on **each** attention head.
>
> Uniform compression rates indeed work well in some cases, but not for all tasks.
> As demonstrated in Table 3 of Appendix D, on certain challenging tasks like CSQA and ARCC, employing a uniform compression rate can sometimes result in a significant drop in accuracy.
> For instance, on ARCC at a 50% cache budget, after applying the greedy search algorithm, the accuracy rises from 34.34 to 36.61.
> This indicates that in practice, the flexible identification of compression levels for different heads may be necessary.
>
>
> **W3 (part 2):** Orthogonal Constraint has less effect when Cache Utilization<0.5.
>
> When orthogonality is not maintained, the model can still learn primary and secondary information and transform the cache features into spaces with reduced dimensions.
>
>
> However, without orthogonality between each column vector within the orthogonal matrices, the space spanned by the cache feature may not be fully utilized due to a lower rank.
> This can lead to a noticeable degradation in performance, especially at higher cache ratios.
>
> To sum up, orthogonality ensures that the full-rank space is effectively utilized, supporting consistent performance across different cache budgets.
>
> [1]:  Zhang et al., "H2O: Heavy-Hitter Oracle for Efficient Generative Inference of Large Language Models", NeurIPS 2023.
>
> [2]: Liu et al., "KIVI: A Tuning-Free Asymmetric 2bit Quantization for KV Cache", ICML 2024.
>
> [3]: Aditya Kusupati, et al. Matryoshka representation learning. Advances in Neural Information Processing Systems, 2022.
>
> [4]: DeepSeek-AI at al., "DeepSeek-V2: A Strong, Economical, and Efficient Mixture-of-Experts Language Model", arXiv 2024.

---

> > ### Comment · Reviewer_76BC · 2024-11-22
> >
> > Thanks for the detailed and careful rebuttal. I appreciate the author's effort. I will keep my score and positive support on the paper.

---

> > > ### Author Response · Authors · 2024-11-22
> > > **Reponse to Reviewer 76BC**
> > >
> > > Thank you for your meticulous review, which has played a pivotal role in improving our work.

---

### Official Review · Reviewer_uec1 · 2024-11-04

**Soundness:** 3
**Presentation:** 3
**Contribution:** 3
**Rating:** 6
**Confidence:** 2

**Summary:**

This paper proposes MatryoshkaKV, a method to compress the key-value (KV) cache in large language models (LLMs) along the feature dimension using trainable orthogonal projection matrices. As LLMs grow in size, the KV cache can become a bottleneck in storage and memory transfer. Previous approaches have focused on compressing the cache along the layer, head, and sequence length dimensions. This work explores compressing along the feature dimension.

The authors first investigate using PCA to obtain orthogonal projection matrices for dimensionality reduction of the keys and values in each attention head. While this works well at moderate compression levels without needing training, performance degrades quickly at higher compression.

To improve on this, they propose MatryoshkaKV which tunes the orthogonal projection matrices end-to-end using a knowledge distillation objective and a special "Matryoshka" training strategy that enables adaptively searching for optimal compression rates per layer and head at inference time. The orthogonality of the projections is enforced using a Cayley parameterization.

**Strengths:**

The paper tackles the problem of KV cache compression in LLMs from a new angle by focusing on the feature dimension. While prior work has explored compressing along the layer, head, and sequence length dimensions, this work shows that significant compression gains can also be achieved along the feature axis. This opens up a promising new direction for efficient LLM inference.

The MatryoshkaKV method demonstrates impressive performance in experiments. It can compress KV caches by 60-75% on average while retaining over 90% of the full model's accuracy. This is a significant improvement over the PCA baseline, especially at high compression rates. The results hold across both continual pre-training and supervised fine-tuning settings, showing the approach is robust and widely applicable.

**Weaknesses:**

The paper lacks rigorous theoretical analysis of why their proposed MatryoshkaKV method works better than PCA-based approaches

While they provide some error analysis in Appendix A, it's relatively brief and doesn't fully explain the theoretical underpinnings of their method's superior performance

Limited evaluation on very long sequence tasks where KV cache compression would be most valuable

**Questions:**

Why does the Matryoshka training strategy work better than static compression ratios? What's the theoretical justification?

How sensitive is the method to the choice of sampling schedule for compression rates during training?

Why is PCA initialization critical for convergence? Could other initialization strategies work?

---

> ### Author Response · Authors · 2024-11-20
> **Response to Reviewer uec1**
>
> We are grateful to the author for providing us with numerous suggestions regarding theory and analysis.
>
> **W1, W2:** The paper lacks rigorous theoretical analysis of why their proposed MatryoshkaKV method works better than PCA-based approaches.
>
> We conduct an in-depth analysis of a single head within a single layer in Appendix A. We find that to achieve the minimum error, it is essential to jointly optimize the projection matrices for both K and V.
> For example, if the principal components of K and V just lie in each other's orthogonal complement, the approximation error would be extremely large.
> This shows that methods like PCA, which do not consider the interaction between K and V, are suboptimal.
> Unfortunately, due to the nonlinear relationship (softmax function) in the calculation between K and V within the attention mechanism, certain mathematical methods such as GSVD (Generalized Singular Value Decomposition) are inapplicable for jointly optimizing the projection matrices of both K and V.
>
> Moreover, the optimal solution also varies with the input data distribution.
> Given the difficulty in modeling the distribution of all corpora globally, we believe that using a data-driven approach for optimization is a reasonable way to minimize the error of the model after KV cache compression on most tasks.
> Thus, we make these orthogonal matrices trainable to obtain optimal results.
> By directly fine-tuning the projection matrices to maximize the data likelihood, we can better adapt to different data distributions and task requirements, thereby enhancing the overall performance of the model.
>
> **Q1:** Why does the Matryoshka training strategy work better than static compression ratios? What's the theoretical justification?
>
> The static strategy has a significant drawback in that it suffers from poor generalization.
> Low-rank projections fail to adapt and perform effectively at compression ratios that are not encountered during the training phase, as proven by results in Figure 4.
> In contrast, our proposed strategy, which **employs randomly sampled compression ratios on each attention head during the training process**, offers several notable advantages:
> - The randomly sampled compression ratios during training endow the low-rank projections with the ability to generalize across all possible compression ratios.
> - Setting various compression ratios across different attention heads effectively decouples the working ratios of different heads. This means that during the testing phase, we have the flexibility to set different ratios for different heads, which aligns with the observed phenomenon of anisotropy in different heads as reported in some related works we've discussed in Section 5.4.
>
> **Q2:** How sensitive is the method to the choice of sampling schedule for compression rates during training?
>
> Thanks for the question. We clarify **our training approach is not sensitive to the selection of the schedule**.
> We have added experiments using other schedules in **Table 4 in General Reply**, where the results do not change significantly.
>
> **Q3:** Why is PCA initialization critical for convergence? Could other initialization strategies work?
>
> Thanks. We identify the importance of PCA initialization empirically.
> We have attempted initialization with *randn_init*, *kaiming_init*, and *xaiver_init*, yet none of these methods have proven to be effective.
> This is due to the introduction of the orthogonality constraint upon the projection matrices, which makes the optimization process probably unstable and suffers from a cold start.
>
> **W3:** Limited evaluation on very long sequence tasks.
>
> **Our approach to KV cache compression is on the feature dim axis rather than the sequence length axis, thus, it applies to both short and long texts**.
> We have supplemented experiments on Longbench to support this view. Our results are displayed in **Table 1 in General Reply**.
>
> Due to GPU memory constraints, we set the maximum length of training samples to 2048 during CPT.
> Accordingly, we truncate LongBench samples to 2048 to calculate perplexity.
> **On average, the perplexity only increases by 0.40, 0.76, and 1.30 at 75%, 50%, and 37.5% KV cache budget**.
> Meanwhile, the experiments of combining our method with H2O can also prove that we can achieve a higher compression ratio for long context, which further demonstrates that **our method can effectively relieve the I/O bottleneck for long context**.
>
> We will add the results of 4K length (the maximum length for LLaMA2-7B-base training samples) in the updated version.

---

### Official Review · Reviewer_XwVW · 2024-11-04

**Soundness:** 3
**Presentation:** 3
**Contribution:** 2
**Rating:** 6
**Confidence:** 3

**Summary:**

The authors study the problem of KV cache compression. While existing work focuses on compression along the layer number, the head number, and the sequence length, the authors work on the feature dimension. While PCA is the most intuitive approach, it does not provide good enough performance. Instead, the authors propose to directly tune the orthogonal projection matrices with a distillation objective using an elaborate Matryoshka training strategy.

**Strengths:**

1. The paper is easy to follow.
2. Much stronger performance than the PCA baseline when the compression ratio is low.

**Weaknesses:**

1. It is unclear whether the novelty of the paper is significant.
2. The paper does not compare with the methods that compress the other dimensions. Thus, it is unclear whether the proposed method is more effective. It is also unclear whether the proposed method can be combined with the others while maintaining its effectiveness.

**Questions:**

1. Can the author provide more insight so that the novelty is more than just the direct application of the Matryoshka training strategy proposed by the other paper?
2. Can the author compare with one to two baselines that compress the other dimensions to show compressing the feature dimension is more effective? Or can the author show that they can be combined together to further improve the performance?

---

> ### Author Response · Authors · 2024-11-20
> **Reponse to Reviewer XwVW**
>
> We sincerely thank the reviewer for the time to read our paper. We are glad you thought our paper was easy to follow.
>
> **Q1 (W1):** It is unclear whether the novelty of paper is significant.
>
> Thanks for the comment. We would like to clarify that solving the KV cache compression problem with the proposed trainable orthogonal projection and Matryoshka training strategy is non-trivial.
> To achieve KV compression, many previous works are based on token eviction and merging. In contrast, we focus on **compression in the feature dimension**, which is completely orthogonal and compatible with them. The corresponding results are displayed in our response to Q2(W2).
>
> By tuning the orthogonal projection with the Matryoshka strategy, we have addressed the issue of training-free PCA to lead to sub-optimal solutions. The Matryoshka strategy ensures the hierarchy of the columns in the orthogonal matrices during training, enabling users to arbitrarily choose KV cache compression ratios for tasks of different difficulties. We have also experimentally demonstrated the importance of this strategy in the Ablation part of Section 5.3.
>
> We kindly ask the reviewer to re-evaluate the contribution of this paper and welcome more specific comments on this.
>
> **Q2 (W2):** Can author show MatryoshkaKV can be combined to further improve performance?
>
> Thanks for the kind suggestion. In particular, the application of MatryoshkaKV to Mistral-v0.3-7B-base in Section 5 can serve as a combination of our method with other dimensions of KV compression because Mistral-v0.3-7B-base uses the Group Query Attention (GQA), which has already compressed the KV in head-number axis.
>
> Additionally, we **combine our methods with H2O, which operates on the sequence axis**, and the results are supplemented in **Table 1 in General Reply**. We see our approach can be effectively combined with H2O, the perplexity on long contexts increases by merely 1.02 at 10% KV cache budget.
> Additionally, if we compress by 50% on both the sequence length and feature dimension axes (with an actual cache usage rate of 25%), we can achieve an average accuracy of 55.85 on 6 benchmarks, which is 91.32% of the baseline.
> This is much better than the effect of only using 25% separately on these two individual axes.
>
> Moreover, the results of the **combination of MatryoshkaKV and KIVI (a KV quantization technique)** are also supplemented in **Table 2 in General Reply**.
>
> The results also reflect our MatryoshkaKV can be used in synergy with KV cache techniques on other dimensions.

---

> > ### Author Response · Authors · 2024-11-25
> > **Sincerely looking forward to the further discussions**
> >
> > Dear reviewer,
> >
> > We kindly inquire if our response has addressed your concerns. If so, we would greatly appreciate your reconsideration of our work and potential score adjustment.
> >
> > If you have any additional questions or suggestions, we would be happy to have further discussions.
> >
> > Best regards,
> >
> > The Authors

---

> ### Author Response · Authors · 2024-11-27
> **Kindly requesting reconsideration**
>
> Dear reviewer,
>
> The revision deadline is approaching, and we sincerely hope to receive any concerns or questions you may have beforehand. If there are any issues, please let us know at your earliest convenience so we can address them promptly.
>
> Best regards,
>
> The Authors

---

> > ### Comment · Reviewer_XwVW · 2024-11-27
> >
> > Hi,
> >
> > I have read the response and I raised my score to 6 due to the experiment showing that MatryoshkaKV can be combined to further improve performance with other compression methods.
> >
> > Sincerely yours,
> > Reviewer XwVW

---

> > > ### Author Response · Authors · 2024-11-29
> > > **Thanks for your feedback**
> > >
> > > We sincerely appreciate your thoughtful feedback and the improved score. Your meticulous review has been invaluable in significantly enhancing the quality of our work.

---

### Official Review · Reviewer_LgQe · 2024-11-04

**Soundness:** 3
**Presentation:** 3
**Contribution:** 3
**Rating:** 6
**Confidence:** 3

**Summary:**

The paper introduces MatryoshkaKV, a method for compressing the Key-Value (KV) cache in large language models (LLMs) to reduce memory during inference. The method begins with PCA for initial dimensionality reduction but addresses PCA’s limitations by tuning projection matrices through knowledge distillation and applying Matryoshka training strategy to enable adaptive compression, allowing the model to balance performance and compression. Furthermore, this paper demonstrates effectiveness with high compression rates while maintaining relatively high accuracy across various LLMs on both CPT and SFT tasks.

**Strengths:**

1. The proposed Matryoshka training strategy effectively preserves hierarchical structures in orthogonal matrices inherited from PCA at various compression levels, ensuring robust performance across dimensions.

2. Greedy search algorithm effectively adapts to differing sparsity in each
$𝑊_𝑘$ and $𝑊_𝑣$ matrix, showcasing flexibility in compression rates across layers.

3. There are comprehensive MKV evaluations across cache budgets, which reveals substantial improvements, particularly under extremely low cache budget.

**Weaknesses:**

1. Lack of Runtime Evaluation: The absence of runtime metrics makes it challenging to assess the practical benefits of this method fully (see Questions).

2. Missing State-of-the-Art Comparisons: Unusually, the paper doesn’t thoroughly compare to existing state-of-the-art methods. Although it mentioned the other methods may collapse under  60% cache budget (lines 126-131), a comparison with Eigen-Attention and HeadKV at different cache budgets and tasks in terms of both performance and runtime would strengthen the evaluation.

**Questions:**

1. Although the paper mentions it only needs processing 2 million training tokens (line 104), it does not clarify the runtime for each base model and task. Please provide the runtime details for the KV compression process, including PCA initialization, the greedy search for compression level selection, and fine-tuning in both CPT and SFT tasks. Notably, since both the greedy search and compression levels rely on outputs from the original model, this could potentially double the training and inference times.

2. How is $\Delta𝑟$ determined in the greedy search algorithm? And what values are used in the experiments?

3. In Fig.4, MKV seems to work well with uniform compression rates. Does this always apply to all tasks?

3. Line 96: k→r?

---

> ### Author Response · Authors · 2024-11-20
> **Reponse to Reviewer LgQe**
>
> We would like to express our sincere gratitude for your careful review and the valuable concern you raised regarding the time consumption of our work. Your feedback has provided us with an opportunity to further clarify and enhance the description of our method.
>
> **W1 (Q1):** Lack of Runtime Evaluation.
>
> We first clarify that in the continual pretraining (CPT) experiments, the process of our approach is (1) obtaining the PCA initialization based on a small subset of a general corpus, (2) training our model on the CPT corpus, (3) searching for the heterogeneous compression levels for various heads with a small calibration dataset (5 - 10 samples) on the specific task of concern, and (4) performing inference on that task given the identified compression levels.
> The time for steps (1-3) can be substantially amortized by multiple downstream tasks, and thus should not be a concern. Besides, the training time for our approach on the CPT corpus is 8 hours using 4 A100 GPUs.
> Regarding the wall-clock time for per inference step, our current implementation consumes a similar time as the baseline full-KV model:
>
> The inference speed of our MatryoshkaKV under the uniform compression rate during the inference process with a batch size of 32.
> |      | LLaMA2-7B-base | 100% | 87.5% | 75% | 62.5% | 50% | 37.5% | 25% |
> |------|------|------|------|------|------|------|------|------|
> | Tokens per second | 33.65 | 34.12 | 34.08 | 34.90 | 35.27 | 36.42 | 36.75 | 37.22 |
>
> We clarify our paper mainly focuses on reducing storage and memory transfer costs instead of runtime, and we have proven that our approach indeed consumes significantly less storage cost than the baseline.
> We will make these points clear in the revised revision.
>
> **W2:** Missing State-of-the-Art Comparisons.
>
> Sorry for the missing comparisons. We first clarify that our PCA baseline is roughly identical to Eigen-Attention [1] (the details are described in Section 3.2) and has a similar performance as Eigen-Attention [1] (see Table 2 in [1] and Table 1 in ours), so we take our PCA baseline as a surrogate of Eigen-Attention [1]. Our existing results have reflected the superiority of our method.
>
> Regarding HeadKV, we first clarify that it has not been open-sourced. It uses low-rank orthogonal matrices and merges multiple heads into one. There are certain similarities between HeadKV and our method: both use orthogonal matrices and have basically the same parameterization. However, the dimensions in which we and HeadKV perform compression are inherently different, and HeadKV also does not further optimize the orthogonal matrices, hence still suffers from suboptimal performance due to the confounding issues. We have added these discussions to the revision and promise to add HeadKV to the empirical comparison once HeadKV is open-sourced.
>
> Besides, we have **compared to another related work, ASVD [2]**, the results are supplemented in **Table 3 in General Reply**.
>
> ASVD notices the low-rank property of LLM parameters and compresses the KV cache while compressing the model parameters. Since it only releases checkpoints for three cache budgets of 85%, 90%, and 95%, we mainly compare our method with them in these three budgets. The results exhibit that our MatyoshkaKV performs better than ASVD at all three cache budget levels.
>
>
> **Q2:** How is $\Delta r$ determined in the greedy search algorithm? And what values are used in experiments?
>
> We simply choose $\Delta r = d / 8$ in the greedy search algorithm, because our predefined compression rate schedule is { $ \frac{i}{8}  (i=1,2, ...,8) $ }, These values can be further tuned for better empirical results, left as future work.
>
> **Q3:** MKV seems to work well with uniform compression rates. Does this always apply to all tasks?
>
> We clarify uniform compression rates do not always work well for all tasks.
> As shown in Table 3 of Appendix D, on some challenging tasks such as CSQA and ARCC, using a uniform compression rate may lead to an accuracy drop of more than 2 percentage points.
> Specifically, at a 50% cache budget on ARCC, after applying the greedy search algorithm, the accuracy increases from 34.34 to 36.61.
> Thus, the flexible identification of the compression levels for various heads can be necessary in practice.
> And, we would like to comment that the greedy search-based identification algorithm can be efficient enough in practice, using a calibration set of only 5 - 10 samples.
>
> **Q4:**: k→r?
>
> We sincerely apologize for this error. We have already corrected it.
>
> [1]: Saxena et al., "Eigen Attention: Attention in Low-Rank Space for KV Cache Compression.", ArXiv 2024.
>
> [2]: Yuan et al., "ASVD: Activation-aware Singular Value Decomposition for Compressing Large Language Models", ArXiv 2024

---

> > ### Author Response · Authors · 2024-11-25
> > **Sincerely looking forward to the further discussions**
> >
> > Dear Reviewer,
> >
> > We hope that our clarifications and detailed explanations of our methods have adequately addressed your concerns. If our responses have resolved your queries, we kindly hope you might reconsider adjusting your score.
> > Should you have any further questions or suggestions, we would be more than happy to engage in additional discussions to improve our work further.
> >
> > Thank you for your time and thoughtful review.
> >
> > Best regards,
> >
> > The Authors

---

> > > ### Comment · Reviewer_LgQe · 2024-11-26
> > > **Score change**
> > >
> > > Thanks for the detailed rebuttal. I appreciate the author's effort. I will raise my score.

---

> > > > ### Author Response · Authors · 2024-11-27
> > > > **Thanks for your feedback**
> > > >
> > > > Thank you for recognizing our work. Your invaluable suggestions have played a crucial role in enhancing the quality of our manuscript.

---

### Author Response · Authors · 2024-11-20
**General Response (Part 4/4)**

- [Schedule Choice @LgQe, @uec1]: In Appendix G, we include experiments that demonstrate how our choice of predefined compression rate schedules during training impacts the results, which are also listed here:

Table 4: Accuracy of our MatryoshkaKV after CPT on six benchmarks. We use uniform compression levels for inference here for simplicity. Different hyper-parameters are compared. In the table we donate  the schedule $ M_{1}  =  \frac{i}{4} d (i = 1,2,3,4) $ , and the schedule $ M_{2}  =  \frac{i}{8} d (i = 1,2, ...,8) $. We use uniform compression levels for inference here for simplicity.
| **Model**         | **Budget** | **Method** | **HLSG** | **ARC-C** | **ARC-E** | **PIQA** | **WG** | **CSQA** | **Avg.** |
|-------------------|------------|------------|----------|-----------|-----------|----------|--------|----------|----------|
| LLaMA2 7B-base    | 100.0%     | $\mathcal{M}_{1}$ | 72.03    | 36.61     | 52.56     | 76.71    | 61.64  | 67.16    | 62.07    |
|                   |            | $\mathcal{M}_{2}$ | 72.05    | 37.29     | 52.38     | 76.66    | 61.72  | 67.32    | 61.24    |
|                   | 87.5%      | $\mathcal{M}_{1}$ | 72.03    | 37.29     | 53.09     | 76.28    | 62.75  | 65.77    | 62.18    |
|                   |            | $\mathcal{M}_{2}$ | 72.22    | 35.93     | 52.20     | 76.28    | 62.12  | 65.27    | 60.67    |
|                   | 75.0%      | $\mathcal{M}_{1}$ | 70.79    | 34.92     | 53.62     | 76.88    | 60.54  | 65.03    | 61.31    |
|                   |            | $\mathcal{M}_{2}$ | 70.98    | 34.58     | 55.20     | 76.77    | 61.56  | 63.64    | 60.46    |
|                   | 62.5%      | $\mathcal{M}_{1}$ | 69.03    | 32.88     | 52.91     | 74.86    | 59.19  | 64.54    | 59.69    |
|                   |            | $\mathcal{M}_{2}$ | 69.22    | 37.29     | 55.73     | 75.22    | 59.35  | 64.21    | 60.17    |
|                   | 50.0%      | $\mathcal{M}_{1}$ | 66.34    | 32.88     | 53.09     | 74.97    | 58.25  | 62.49    | 58.59    |
|                   |            | $\mathcal{M}_{2}$ | 66.62    | 34.24     | 52.91     | 75.46    | 58.41  | 62.00    | 58.27    |
|                   | 37.5%      | $\mathcal{M}_{1}$ | 61.55    | 31.19     | 49.91     | 73.83    | 56.27  | 52.09    | 53.78    |
|                   |            | $\mathcal{M}_{2}$ | 62.38    | 32.20     | 50.26     | 73.34    | 56.67  | 55.28    | 55.02    |
|                   | 25.0%      | $\mathcal{M}_{1}$ | 50.91    | 26.10     | 44.97     | 68.39    | 52.72  | 38.33    | 46.38    |
|                   |            | $\mathcal{M}_{2}$ | 51.91    | 27.46     | 44.44     | 69.64    | 54.54  | 44.39    | 48.73    |

- [Inference speed @LgQe, @9USD]: We evaluate the inference speed of our LLM equipped with MatryoshkaKV. During the inference process with a batch size of 32, our current implementation consumes a slightly faster time than the baseline full-KV model.

Table 5: The inference speed of our MatryoshkaKV under the uniform compression rate during the inference process with a batch size of 32.
|      | LLaMA2-7B-base | 100% | 87.5% | 75% | 62.5% | 50% | 37.5% | 25% |
|------|------|------|------|------|------|------|------|------|
| Tokens per second | 33.65 | 34.12 | 34.08 | 34.90 | 35.27 | 36.42 | 36.75 | 37.22 |

[1]: Saxena et al., "Eigen Attention: Attention in Low-Rank Space for KV Cache Compression.", ArXiv 2024.

[2]:  Zhang et al., "H2O: Heavy-Hitter Oracle for Efficient Generative Inference of Large Language Models", NeurIPS 2023.

[3]: Liu et al., "KIVI: A Tuning-Free Asymmetric 2bit Quantization for KV Cache", ICML 2024.

[4]: Yuan et al., "ASVD: Activation-aware Singular Value Decomposition for Compressing Large Language Models", ArXiv 2024

---

### Author Response · Authors · 2024-11-20
**General Response (Part 3/4)**

- [Combination with KIVI @76BC, @XwVW, @9USD]: In Appendix E, we add the results of the combination of our method with KIVI [3], also presented below:


Table 2: Results of Combination of Distilled MatryoshkaKV Projections and KIVI (2bit KV cache quantization) on Six Benchmarks. We use uniform compression levels for inference here for simplicity.
| **Model**         | **Budget** | **Method** | **HLSG** | **ARC-C** | **ARC-E** | **PIQA** | **WG**  | **CSQA** | **Avg.** |
|-------------------|------------|------------|----------|-----------|-----------|----------|---------|----------|----------|
| **LLaMA2 7B-base** | 100.0%     | MKV        | 70.89    | 36.95     | 53.26     | 76.39    | 61.56   | 67.08    | 60.98    |
|                   |            | MKV+KIVI   | 69.76    | 35.93     | 51.98     | 76.55    | 61.48   | 66.26    | 60.49    |
|                   | 87.5%      | MKV        | 70.87    | 36.95     | 51.15     | 76.17    | 61.80   | 64.95    | 60.47    |
|                   |            | MKV+KIVI   | 70.45    | 36.61     | 50.79     | 76.44    | 61.01   | 64.13    | 59.91    |
|                   | 75.0%      | MKV        | 69.30    | 33.90     | 54.67     | 75.90    | 61.09   | 63.23    | 60.08    |
|                   |            | MKV+KIVI   | 68.62    | 32.20     | 54.85     | 76.06    | 60.30   | 63.23    | 59.68    |
|                   | 62.5%      | MKV        | 67.25    | 36.27     | 53.62     | 75.52    | 59.27   | 64.46    | 59.39    |
|                   |            | MKV+KIVI   | 66.56    | 35.25     | 51.68     | 75.41    | 59.43   | 61.59    | 58.33    |
|                   | 50.0%      | MKV        | 65.08    | 33.56     | 52.03     | 74.81    | 57.54   | 60.36    | 56.98    |
|                   |            | MKV+KIVI   | 63.25    | 32.54     | 51.15     | 74.43    | 57.38   | 59.46    | 56.35    |
|                   | 37.5%      | MKV        | 61.02    | 29.83     | 49.21     | 73.45    | 55.64   | 55.36    | 54.09    |
|                   |            | MKV+KIVI   | 57.11    | 28.81     | 48.85     | 71.71    | 55.64   | 50.37    | 52.08    |
|                   | 25.0%      | MKV        | 50.61    | 25.76     | 45.33     | 69.64    | 54.30   | 43.90    | 47.96    |
|                   |            | MKV+KIVI   | 48.12    | 27.80     | 42.86     | 67.85    | 53.59   | 40.54    | 46.78    |

The results show that our MatryohskaKV can be easily combined with KV quantization techniques and achieve a higher compression rate.

- [Baseline @LgQe, @9USD]: In Appendix F, we add some baselines such as ASVD [4] and compare with our MatryoshkaKV， with the results listed below:

Table 3: Comparison between our MatryoshkaKV and baseline ASVD. We use uniform compression levels for inference here for simplicity.
| **Model**              | **Budget** | **Method** | **HLSG** | **ARC-C** | **ARC-E** | **PIQA** | **WG** | **CSQA** | **Avg.** |
|-----------------------|------------|------------|----------|-----------|-----------|----------|--------|----------|----------|
| LLaMA2                | 100.0%     | baseline   | 74.00    | 35.93     | 50.97     | 78.50    | 61.64  | 65.93    | 61.16    |
|                       | 95%        | ASVD       | 71.12    | 36.95     | 52.20     | 76.28    | 62.35  | 66.67    | 60.92    |
|                       |            | MKV        | 72.59    | 36.27     | 53.09     | 76.44    | 62.43  | 66.75    | 61.25    |
|                       | 90%        | ASVD       | 70.45    | 34.92     | 52.03     | 75.63    | 61.72  | 64.70    | 60.06    |
|                       |            | MKV        | 72.30    | 36.61     | 54.50     | 76.50    | 62.90  | 65.93    | 62.03    |
|                       | 85%        | ASVD       | 67.23    | 35.93     | 50.26     | 74.86    | 60.38  | 62.16    | 59.29    |
|                       |            | MKV        | 72.33    | 35.93     | 53.26     | 76.33    | 61.80  | 64.78    | 61.13    |

---

### Author Response · Authors · 2024-11-20
**General Response (Part 2/4)**

**Simultaneously, we add some experiments requested by the reviewers:**
- [Combination with H2O @76BC, @uec1, @XwVW, @9USD]: In Appendix E, we add the results of the combination of our method with H20 [2] and show perplexity on LongBench, which are also listed below.

Table 1: Results of Combination of Distilled MatryoshkaKV Projections and H2O across Seven Benchmarks. We use uniform compression levels for inference here for simplicity. The first and second columns indicate the individual compression rates along two axes. If H2O uses 20% cache on the sequence length axis and MatryoshkaKV uses 50% cache on the feature dimension axis, the overall cache utilization is 10%.
| **H₂O** | **MKV** | **LongBench** | **HLSG** | **ARC-C** | **ARC-E** | **PIQA** | **WG** | **CSQA** | **Avg.** |
|---------|---------|---------------|----------|-----------|-----------|----------|--------|----------|----------|
| 100%    | 100%    | 4.17          | 72.05    | 37.29     | 52.38     | 76.66    | 61.72  | 67.32    | 61.24    |
|         | 87.5%   | 4.44          | 72.22    | 35.93     | 52.20     | 76.28    | 62.12  | 65.27    | 60.67    |
|         | 75.0%   | 4.57          | 70.98    | 34.58     | 55.20     | 76.77    | 61.56  | 63.64    | 60.46    |
|         | 62.5%   | 4.70          | 69.22    | 37.29     | 55.73     | 75.22    | 59.35  | 64.21    | 60.17    |
|         | 50.0%   | 4.93          | 66.62    | 34.24     | 52.91     | 75.46    | 58.41  | 62.00    | 58.27    |
|         | 37.5%   | 5.47          | 62.38    | 32.20     | 50.26     | 73.34    | 56.67  | 55.28    | 55.02    |
|         | 25.0%   | 7.66          | 51.91    | 27.46     | 44.44     | 69.64    | 54.54  | 44.39    | 48.73    |
| 75%     | 100%    | 4.18          | 70.71    | 36.61     | 52.38     | 76.55    | 60.54  | 66.50    | 60.55    |
|         | 87.5%   | 4.44          | 71.42    | 35.25     | 53.09     | 76.33    | 59.91  | 64.62    | 60.74    |
|         | 75.0%   | 4.57          | 70.31    | 34.34     | 54.14     | 76.39    | 59.27  | 62.90    | 59.94    |
|         | 62.5%   | 4.70          | 68.47    | 36.27     | 54.32     | 75.41    | 58.48  | 63.96    | 59.89    |
|         | 50.0%   | 4.94          | 66.00    | 32.54     | 51.50     | 75.63    | 57.30  | 61.43    | 57.46    |
|         | 37.5%   | 5.47          | 61.50    | 32.88     | 49.21     | 73.01    | 55.09  | 55.12    | 54.63    |
|         | 25.0%   | 7.67          | 51.32    | 27.80     | 44.09     | 69.37    | 53.59  | 44.55    | 48.47    |
| 50%     | 100%    | 4.20          | 68.72    | 33.22     | 52.20     | 76.12    | 56.67  | 64.78    | 58.62    |
|         | 87.5%   | 4.46          | 67.89    | 34.58     | 51.85     | 76.28    | 55.88  | 62.00    | 58.13    |
|         | 75.0%   | 4.59          | 66.01    | 35.59     | 53.79     | 75.41    | 54.54  | 62.00    | 58.05    |
|         | 62.5%   | 4.73          | 63.59    | 34.92     | 51.32     | 75.68    | 55.25  | 60.52    | 57.04    |
|         | 50.0%   | 4.96          | 61.33    | 36.10     | 50.74     | 73.67    | 55.57  | 57.67    | 55.85    |
|         | 37.5%   | 5.50          | 59.26    | 29.83     | 49.91     | 73.61    | 53.04  | 54.14    | 53.29    |
|         | 25.0%   | 7.71          | 49.44    | 26.44     | 41.80     | 68.72    | 52.96  | 43.24    | 46.94    |
| 20%     | 100%    | 4.40          | 61.55    | 25.76     | 41.27     | 73.29    | 53.28  | 47.01    | 49.98    |
|         | 87.5%   | 4.65          | 61.36    | 30.51     | 39.86     | 73.72    | 52.09  | 49.06    | 50.94    |
|         | 75.0%   | 4.79          | 60.29    | 28.47     | 38.62     | 72.75    | 53.12  | 50.45    | 50.62    |
|         | 62.5%   | 4.93          | 58.77    | 26.78     | 39.86     | 70.84    | 52.72  | 49.30    | 50.58    |
|         | 50.0%   | 5.19          | 56.39    | 26.78     | 38.10     | 71.22    | 51.62  | 49.16    | 49.66    |
|         | 37.5%   | 5.74          | 52.12    | 23.39     | 34.22     | 68.50    | 52.17  | 41.44    | 44.82    |
|         | 25.0%   | 8.01          | 43.22    | 21.02     | 31.92     | 63.93    | 51.38  | 33.09    | 40.75    |


According to the results, by concurrently using MatryoshkaKV and H2O, the perplexity on long contexts increases by merely 1.02 at 10% KV cache budget.
Additionally, if we compress by 50% on both the sequence length and feature dimension axes (with an actual cache usage rate of 25%), we can achieve an average accuracy of 55.85 on 6 benchmarks, which is 91.32% of the baseline.
This is much better than the effect of only using 25% separately on these two individual axes.

---

### Author Response · Authors · 2024-11-20
**General Response (Part 1/4)**

We thank the reviewers for their thoughtful feedback. We are encouraged that the reviewers found our MatryoshkaKV has comprehensive evaluations (@LgQe, @76BC) and robust performance (@LgQe, @XwVW, @uec1, @9USD), being flexible (@LgQe, @XwVW, @76BC) and novel for KV cache compression (@uec1, @76BC).
**We have made a revision of our previous paper and highlighted the modified parts in red**. The following are the modifications we have made to our paper:

In the experiment part of Section 5, specifically in the subsection of CPT, **we add numerous experimental details**, including:

- [Clarification @9USD]: The training dataset used in CPT and its source.
- [Consumption @LgQe]: The runtime metrics for the experimental part.
- [Clarification @LgQe]: The $\Delta r$ used in the greedy search for adaptive compression rates before the inference stage.

**We provide a more detailed explanation of our method:**
- [Clarification @LgQe]: In Section 4, we clarify the whole pipeline of our method.
- [Analysis @uec1]: In Appendix A, we refine some mathematical formulae.

---

### Meta-Review · Area_Chair_kKYk · 2024-12-18

**Metareview:**

Summary:
MatryoshkaKV introduces a novel method for compressing Key-Value (KV) cache in large language models along the feature dimension using trainable orthogonal projection matrices. The method combines PCA initialization with knowledge distillation and a Matryoshka training strategy to enable adaptive compression rates across different layers and attention heads. The approach achieves significant compression (up to 95%) while maintaining over 90% of the original model's accuracy.

Main Strengths:

- Novel approach to KV cache compression focusing on the previously unexplored feature dimension
- Strong empirical results showing significant compression with minimal performance degradation
- Compatibility with other compression techniques (demonstrated with GQA, H2O, and KIVI)
- Practical implementation requiring minimal hyperparameter tuning
- Comprehensive ablation studies validating key components

Main Weaknesses:

- Limited theoretical analysis explaining why the method outperforms PCA-based approaches
- Runtime evaluation could be more comprehensive
- Current implementation of heterogeneous ranks may not provide runtime benefits without system-level optimizations
- Task-specific calibration requirement for optimal compression rates

**Additional Comments On Reviewer Discussion:**

Outcomes from Author-Reviewer Discussion:
The authors have addressed several key concerns through their responses:

- Provided runtime metrics showing comparable inference speed to baseline
- Demonstrated compatibility with other compression methods
- Clarified the calibration process (5-10 samples per task)
- Added comparisons with additional baselines (ASVD)
- Explained the practical implications of heterogeneous compression rates

Reviewer Agreement/Disagreement:
Initial ratings ranged from 5 to 8, with most reviewers increasing their scores after author responses. Consensus emerged around accepting the paper, acknowledging its practical value despite some theoretical limitations.

Suggestions for Improvement:

- Strengthen theoretical analysis of why the method outperforms PCA
- Add system-level optimizations for heterogeneous compression
- Include more comprehensive runtime evaluations
- Clarify the calibration process and its practical implications
- Consider additional experiments with newer model architectures

---

### Decision · Program_Chairs · 2025-01-22

Accept (Poster)